# When Are Concepts Erased From Diffusion Models?

Kevin Lu[1]     Nicky Kriplani[2]     Rohit Gandikota[1]     Minh Pham[2]     David Bau[1]

Chinmay Hegde[2]                    Niv Cohen[2]

[1]Northeastern University
[2]New York University

## Abstract

In concept erasure, a model is modified to selectively prevent it from generating a target concept. Despite the rapid development of new methods, it remains unclear how thoroughly these approaches remove the target concept from the model. We begin by proposing two conceptual models for the erasure mechanism in diffusion models: (i) interfering with the model's internal guidance processes, and (ii) reducing the unconditional likelihood of generating the target concept, potentially removing it entirely. To assess whether a concept has been truly erased from the model, we introduce a comprehensive suite of independent probing techniques: supplying visual context, modifying the diffusion trajectory, applying classifier guidance, and analyzing the model's alternative generations that emerge in place of the erased concept. Our results shed light on the value of exploring concept erasure robustness outside of adversarial text inputs, and emphasize the importance of comprehensive evaluations for erasure in diffusion models[1].

## 1  Introduction

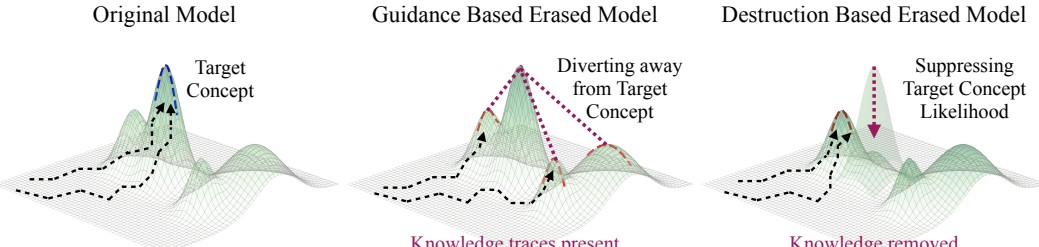

Figure 1: We suggest that diffusion model concept erasure methods can be broadly categorized into two types: (1) Guidance-Based Avoidance, which avoids a concept by redirecting the model to different concept locations. (2) Destruction-Based Removal, which reduces the unconditional likelihood of the target concept while keeping guidance intact, forcing the model to another concept when prompted with the target concept. The height represents the unconditional likelihood $P(X)$.

When a concept is supposedly erased from a diffusion model, is its knowledge truly removed? Or is the model merely avoiding the concept, with the underlying knowledge still intact? This fundamental question is at the heart of understanding erasure in diffusion models.

---

[1]Source code and datasets can be found at kevinlu4588/WhenAreConceptsErased.

39th Conference on Neural Information Processing Systems (NeurIPS 2025).

Investigating this question is crucial as it directly impacts how we evaluate the thoroughness of unlearning methods in text-to-image diffusion models, pushing us towards more rigorous standards. Furthermore, a clear understanding of the underlying erasure mechanisms will enable us to advance research, paving the way for stronger and more robust unlearning techniques.

To help analyze how erasure might affect the underlying model's knowledge, we propose two conceptual mechanisms for erasure methods: *guidance-based avoidance* and *destruction-based removal*. Guidance-based avoidance suggests that the model learns to steer its generation away from the target concept by modifying the conditional guidance, which may leave the core knowledge preserved. In contrast, destruction-based removal implies that the process aims to fundamentally suppress, or ideally, eliminate the underlying knowledge about the concept (see Figure 1).

Distinguishing between these regimes is not straightforward: both can appear visually successful when tested with standard prompts. Prior research [18, 26, 29, 21] has shown that erased concepts can be resurfaced by searching for the right input, suggesting that most existing methods act through guidance-based avoidance rather than destruction-based removal. Yet these findings, while revealing, leave open a question: if the underlying knowledge persists, through which other methods might we uncover it? Can it re-emerge through other techniques as well?

We suggest a multi-perspective approach for testing for persistent knowledge. First, we employ existing input optimization techniques, including textual inversion and prompt-based adversarial attacks, to actively search for inputs that might still trigger the generation of the erased concept. Second, we use context-based probing, where the model is provided with contextual cues related to the erased concept. For example, through inpainting tasks or by initiating the diffusion process from an intermediate step. Conditioned on such context, we see if the edited model can complete or generate the concept. Third, we explore training-free trajectory expansion, which broadens the model's standard diffusion pathways to potentially uncover latent or suppressed concept representations. Fourth, we employ latent classifier guidance, augmenting text conditioning with gradients from a concept-specific latent classifier. This provides a powerful signal that steers the diffusion trajectory toward residual concept latents, countering erasure-induced guidance that drives the model away. Finally, our suite includes dynamic concept tracing to monitor how a concept's representation and its likelihood of generation evolve throughout the entire erasure procedure.

Our findings reveal undiscovered behavior of models under these new evaluation contexts. For instance, models that appear robust under traditional input search techniques remain vulnerable when assessed from other perspectives. These observations emphasize the critical need for a comprehensive suite of evaluations, like the one we propose, to reliably assess the completeness and true effectiveness of any concept erasure method.

## 2 Two Conceptual Models for Erasure

Here, we formalize two conceptual models that capture how diffusion model erasure methods can modify the generative process: guidance-based avoidance and destruction-based removal. While recent works have proposed numerous erasure algorithms (see Section 4 for a review), they are typically described by their training losses, data requirements, or parameter modifications [8, 26, 19, 10]. Often, less attention is given to their effect on the resulting model behavior. Here, we instead focus on their functional effect on the model's output distribution.

We term methods that redirect the model's conditional guidance rather than eliminate the concept itself as *guidance-based* approaches. Accordingly, such approaches may still regenerate the erased concept when given optimized inputs or alternative cues. *Destruction-based* approaches, in contrast, aim to suppress the model's unconditional likelihood $P(X)$ of producing the erased concept. Such methods correspond to a deeper removal of underlying features (and potentially broader collateral effects on nearby concepts).

For example, we examine *Unified Concept Erasure (UCE)* [10] as a representative case of guidance-based erasure. UCE optimizes a new attention projection matrix $W$ using the following loss:

$$\mathcal{L}(W) = \sum_{c_i \in E} \|Wc_i - v_i^*\|^2 + \sum_{c_j \in P} \|Wc_j - W_{\text{old}}c_j\|^2, \tag{1}$$

where $E$ and $P$ denote erased and protected concepts, respectively. The first term aligns erased concept embeddings $c_i$ with neutral or substitute vectors $v_i^*$, while the second term preserves responses

for protected prompts $c_j$. This effectively shifts the model's conditional distribution from $P(X \mid c_i)$ to $P(X \mid c_i^*)$, where $c_i^*$ represents a neutralized prompt (e.g., an empty string).

Because the erased concept's representation is redefined through a substitute projection, UCE alters the semantic associations between text tokens and their corresponding visual features rather than eliminating the features themselves. The edited projection matrix $W$ effectively redirects the model's conditional mapping from the erased concept toward a neutral or generic concept. As a result, the model's output distribution $P(X)$ exhibits **redirection** rather than complete **suppression**: the conditional generation shifts from the erased concept's toward a different one. Therefore, UCE exemplifies a *guidance-based* approach - the underlying concept remains present but the generation is guided away from it under standard prompts.

However, for methods whose post-erasure behavior is less interpretable, such as *STEREO* [26], it becomes less clear whether suppression arises from guidance redirection or true knowledge destruction. To empirically differentiate these regimes, we next introduce a comprehensive evaluation framework that probes multiple pathways through which erased knowledge could resurface. By using many different probing techniques and checking which can still recover the erased concept, we assess the extent and character of the model's residual knowledge, and whether an erasure method behaves more like guidance-based avoidance or destruction-based removal.

# 3 Evaluation Suite

We present our evaluation suite and apply it to a representative set of existing erasure methods. We choose methods that represent different approaches, but our evaluations can be easily applied to any new or existing method. Specifically, we evaluate the following erasure methods:

*Baseline* [20] - Unedited Stable Diffusion 1.4 model (no erasure); *UCE* [10] - A closed-form solution editing of the cross-attention weight in the model to replace the target concept and preserve other concepts; *ESD-u* [8] - fine-tunes the pre-trained diffusion U-Net model weights to remove a specific style or concept when conditioned on a specific prompt; *ESD-x* [8] - fine-tunes only the cross-attention layers, modifying how textual conditioning influences latent feature modulation; *Task Vector* [19] - Finetuning the U-net to increase the likelihood of the target concept, and then editing the model in the opposite direction using the Task Vector technique [15]; *GA* - direct gradient ascent to reduce the likelihood of the target concept; *STEREO* [26] - A two-stage method combining adversarial prompt search with compositional fine-tuning to robustly erase concepts while preserving model utility; *RECE* [12] - A fast, closed-form method that iteratively applies UCE on text embeddings while minimizing impact on unrelated concepts.

*Concepts* - we conduct our experiments on 10 object concepts and 3 art styles. We report average results in the main text, and standard deviation in the supplementary materials. *Metrics* - CLIP: we evaluate semantic similarity of the output image to the target concept name [14]; Classification Accuracy: We detect the presence of the concept in the generated image using an ImageNet classifier for object concepts. For the specificity of model erasure, we measure how the erasure affects other unrelated concepts via CLIP and classification scores). Please see App.C for all model training, erasing evaluation, and metric calculation implementation details.

We refer to each of the following tests as a distinct **probe**: designed to challenge the examined erasure method and reveal the underlying behavior of different methods.

## 3.1 Optimization-based Probing

**Question 1:** Can we probe out the residual knowledge by searching for the right input?

We evaluate this question by adopting strategies from previous works [18, 32]. These methods optimize the inputs of the erased model to search for the right trigger that would resurface the knowledge of the erased concept, if still present. To this end, we use Textual Inversion [6] and an adversarial attack, UnlearnDiffAtk [32]. Both these methods optimize the text embeddings or tokens to generate the erased concept using the erased model. We use them as probes to quantify if there are traces of knowledge present post-erasure, as done in prior work [18].

The results in Table 1 reveal a stark dichotomy of how various methods withstand optimization-based probes. GA, TV, and STEREO exhibit thorough removal of the erased concept, as indicated by the lower CLIP similarity and classification accuracies across both probes. In contrast, methods such as UCE and ESD-x remain highly vulnerable to both Textual Inversion and UnlearnDiffAtk, with high classification accuracy and CLIP scores, suggesting that residual knowledge of the target concept persists. Moreover, a consistent trend emerges: models that are more robust to adversarial probing often suffer greater degradation in their performance on unrelated concepts (see Fig. 8 in App C.3.3).

|  | GA | UCE | ESD-x | ESD-u | TaskVec | STEREO | RECE |
|---|---|---|---|---|---|---|---|
| **Erased Concept (↓)** | | | | | | | |
| CLIP | 24.3 | 22.4 | 21.1 | 20.9 | 23.1 | **19.6** | 21.15 |
| Class Acc. (%) | **0.6** | 4.4 | 3.6 | 1.0 | 2.2 | 0.0 | 4.0 |
| **Textual Inversion (↓)** | | | | | | | |
| CLIP | **22.7** | 30.7 | 30.6 | 28.0 | 25.1 | 24.5 | 29.15 |
| Class Acc. (%) | **0.6** | 71.2 | 65.9 | 31.8 | 6.2 | 6.3 | 58.20 |
| **UnlearnDiffAtk (↓)** | | | | | | | |
| CLIP | **26.0** | 28.3 | 28.7 | 27.8 | 27.1 | 26.1 | 27.9 |
| Class Acc. (%) | 6.5 | 26.8 | 21.0 | 16.6 | 10.3 | **3.7** | 7.2 |
| **Unrelated Concepts (↑)** | | | | | | | |
| CLIP | 28.8 | **31.2** | 30.8 | 30.7 | 29.4 | 0.0 | 30.5 |
| Class Acc. (%) | 52.2 | **75.0** | 71.3 | 70.4 | 60.4 | 52.8 | 71.7 |

Table 1: Optimization-based probing of residual concept knowledge across erasure methods. We evaluate whether erased concepts can be resurfaced using standard prompts, Textual Inversion, and the UnlearnDiffAtk adversarial attack. The final row shows performance on unrelated concepts (↑), measuring the preservation of general model utility.

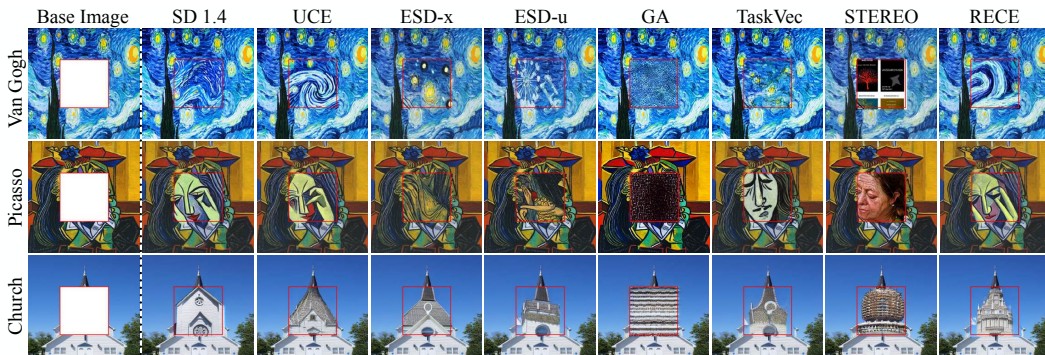

Figure 2: Inpainting-based probe results for multiple erased concepts. For each method and concept, the masked region is filled by the model conditioned on surrounding context. Task Vectors successfully reconstructs the erased region, despite robustness to Textual Inversion and UnlearnDiffAtk.

## 3.2 In-context Probing

**Question 2:** Can we probe out the residual knowledge by providing visual context?

We investigate whether an erased concept can resurface when the model is provided with a single visual in-context example. This question drives our application of visual in-context cues to evaluate the depth and thoroughness of concept erasure (a technique also explored in prior works [1, 23]). Unlike optimization-based approaches, these method do not use the networks gradients, thereby providing a different lens on erasure efficacy.

First, we use *Inpainting* as an in-context probe for erasure. We provide the model with an image corresponding to the concept, but with a portion of image masked out. When the model truly has

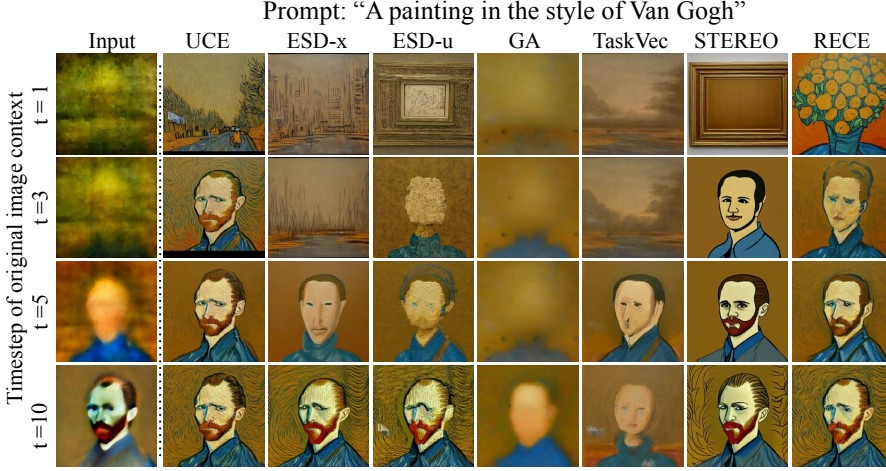

Figure 3: Diffusion Completion outputs given intermediate images generated at timestep $t$ by the original (unerased) model. These noisy inputs are visualized in the first column via the Denoising Trajectory (DT) [7] technique. We then pass each of these unfinished images as contextual inputs to the erased models to complete the remaining denoising steps.

no knowledge of the concept, we assume it should not correctly inpaint the image. Figure 2 shows that the Task Vector method, for example, despite being robust even against Textual Inversion, still inpaints recognizable images of Starry Night by Van Gogh. This is reflected in Table 2, where TaskVectors achieve inpainting CLIP scores comparable to less adversarially robust models. In contrast, STEREO and GA produce little to no meaningful inpainting.

Next, we use *Diffusion Completion* as another in-context probe by leveraging unfinished image generations from the unerased base model. Specifically, we run the diffusion process with the base model for $t = 5$ or $t = 10$ timesteps (out of 50) and save the intermediate image. This image is then passed to the erased model to complete the generation. This approach allows us to quantify how easily the erased concept can be recovered from partial generation traces. Figure 3 shows that RECE and STEREO, despite offering significantly stronger robustness to adversarial attacks compared to methods like UCE, ESD-x, and ESD-u, surprisingly reproduce knowledge about the erased concept during *Diffusion Completion* at $t = 5$ and $t = 10$ respectively.

Given their performance under optimization-based probes, models such as Task Vector, STEREO, and RECE, appeared to have significantly destroyed traces of erased concepts. However, these context-based probing methods reveal a more nuanced model behavior. They offer a complementary perspective to traditional prompt optimization approaches, suggesting that erasure robustness can depend on the nature of the input signal.

| Metric | Base | UCE | ESD-x | ESD-u | GA | TaskVec | STEREO | RECE |
|---|---|---|---|---|---|---|---|---|
| **Inpainting** ($\downarrow$) | | | | | | | | |
| CLIP (Inside mask) | 29.5 | 26.9 | 26.8 | 23.9 | 24.8 | 25.9 | **22.7** | 26.3 |
| Class Acc. (%) | 77.7 | 69.1 | 69.1 | 68.5 | **61.7** | 66.8 | 63.8 | 68.2 |
| **Diffusion Completion** ($\downarrow$) | | | | | | | | |
| CLIP $t=5$ | 30.2 | 27.7 | 27.2 | 26.9 | 24.0 | **23.8** | 23.9 | 28.82 |
| CLIP $t=10$ | 30.2 | 29.6 | 28.7 | 27.5 | **24.5** | 24.9 | 27.8 | 28.82 |
| Class Acc. (%) $t=5$ | 78.0 | 42.7 | 37.8 | 32.5 | **1.1** | 2.4 | 3.2 | 36.5 |
| Class Acc. (%) $t=10$ | 78.0 | 62.1 | 54.8 | 36.9 | **3.2** | 6.1 | 21.2 | 45.4 |

Table 2: *Inpainting* metrics evaluate how well the model completes a masked region when given surrounding context from an image of the erased concept; CLIP scores reflect semantic similarity within the masked area, while classification accuracy considers the full image. *Diffusion Completion* metrics evaluate whether erased concepts resurface when the erased model completes the diffusion process starting from an intermediate image produced by the original model after 5 or 10 out of 50 denoising steps.

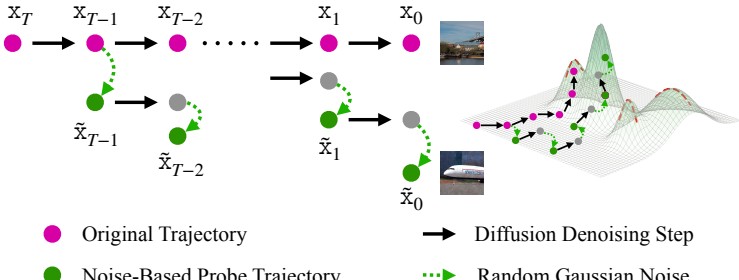

Original Trajectory    Diffusion Denoising Step

Noise-Based Probe Trajectory    Random Gaussian Noise

Figure 4: Our Noise-Based probing technique adds additional noise to the diffusion trajectory. At every diffusion denoising timestep, we add back a controlled amount of noise to allow the model to search in a larger latent space.

## 3.3 Training-Free Trajectory Probing

**Question 3:** Can we probe out the residual knowledge by modifying diffusion trajectory?

We introduce Noise-Based probing, a method to probe for residual knowledge by directly manipulating the model's generation process. This technique searches for hidden knowledge traces by augmenting the diffusion trajectory. Namely, we allow the model to explore alternative generation pathways by simply add Gaussian noise to the intermediate latents after each denoising step:

$$\tilde{x}_{t-1} = (\tilde{x}_t - \alpha\epsilon_D) + \eta\epsilon \tag{2}$$

where $\alpha\epsilon_D$ represents the standard denoising process, and $\eta\epsilon$ represents additional Gaussian noise scaled by parameter $\eta$ (we explore seven values of $\eta$ in the range $[1.0, 1.85]$; see App. D.3). To account for this change and still generate high-quality images, we also scale the diffusion process scheduler variance by $\eta$ and motivate this probing method based on the DDIM formulation in Appendix B.

As illustrated in Fig. 4, this approach performs a controlled exploration of the model's latent space through Brownian motion along the diffusion trajectory. If an edited model's trajectory simply deviates away from a concept, our noise-probe may help to expand the diffusion trajectory bandwidth and find again the better (higher likelihood) images of associated concepts. In other words, the injected noise may enable the model to surface concepts it otherwise suppresses. We emphasize this probe does not optimize an adversarial input or present visual cues, and therefore offers an independent perspective into the internal knowledge of the model. Due to the random nature of this probe, we execute it multiple times and select the generated image with the highest CLIP/classification score relative to the target concept.

Surprisingly, this simple method can reveal traces of knowledge in cases where even powerful optimization-based methods fail. E.g., while adversarially optimized probes fail to recover erased concepts in models like GA and STEREO, the noise-based probe applied to the same prompt and seed successfully restores them (Fig.5). Table 3 shows that Gradient Ascent and STEREO exhibit the strongest robustness against noise-based probing.

| | GA | UCE | ESD-x | ESD-u | TaskVec | STEREO | RECE |
|---|---|---|---|---|---|---|---|
| **Noise-Based Probing** ($\downarrow$) | | | | | | | |
| CLIP | 26.1 | 27.8 | 28.0 | 27.7 | 26.5 | **24.6** | 27.0 |
| Class Acc. (%) | 2.67 | 21.9 | 30.7 | 27.7 | 11.0 | **1.1** | 13.0 |

Table 3: Noise-based probing results. Remarkably, simply increasing the stochasticity of the diffusion process, while keeping the same prompt and seed, can bring erased concepts back. This effect is particularly pronounced for UCE, ESD-x, and ESD-u, indicating that these methods retain recoverable latent traces of the erased concept.

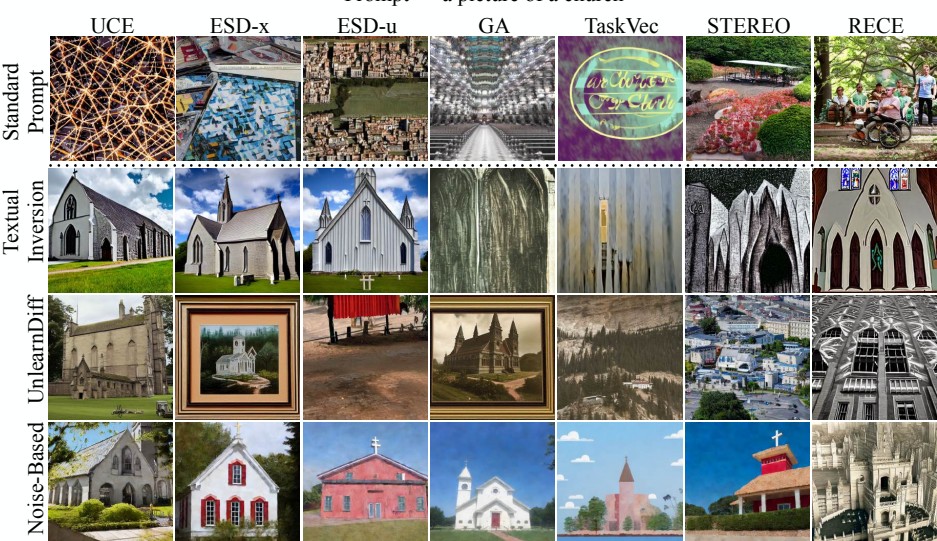

Figure 5: An overview of erasing model behavior under adversarial probes and the Noise-Based probe. Our Noise-Based probe can recover the target concept ("church") even in cases where Textual Inversion and UnlearnDiffAtk fail.

## 3.4 Steered Latent Probing

**Question 4:** Can we probe out the residual knowledge using by classifier guidance?

To probe whether erased models still encode latent traces of the target concept, we apply a variant of classifier guidance in latent space [4]. Instead of relying solely on text conditioning, we train a lightweight classifier directly in latent space (see Appendix D.5 for details). This classifier provides a gradient signal that steers the diffusion trajectory toward regions of latent space associated with the erased concept.

At each denoising step $t$, the current latent $\mathbf{x}_t$ is passed to a timestep-aware classifier $f_{c^*}(\mathbf{x}_t, t)$ trained to detect the target concept $c^*$. For each latent, the classifier outputs a probability $f_{c^*}(\mathbf{x}_t, t) \in [0, 1]$ indicating the presence of the concept. To obtain a semantic direction that increases classifier confidence in the concept, we compute the gradient of the binary cross-entropy loss with respect to the latent, using a target label $y{=}1$ that denotes the concept's presence:

$$\mathbf{g}_t = \nabla_{\mathbf{x}_t} \mathcal{L}_{\mathrm{BCE}}\big(f_{c^*}(\mathbf{x}_t, t), y{=}1\big), \qquad (3)$$

This gradient defines a local direction in latent space that points toward regions the classifier identifies as belonging to the erased concept.

We use the obtained gradient (Eq. 3) to update the latent at each timestep:

$$\tilde{\boldsymbol{x}}_t = \boldsymbol{x}_t - s_{\mathrm{clf}}\, \sigma_t\, \mathbf{g}_t, \qquad (4)$$

where $s_{\mathrm{clf}}$ is the guidance strength and $\sigma_t$ controls the effective step size according to the current noise level (following Dhariwal and Nichol [4]).

During inference, we sweep over $24$ values of $s_{\mathrm{clf}}$ and select the sample with the highest classification score for the target concept. Interestingly, models trained with STEREO, which are robust to Textual Inversion and UnlearnDiffAtk, can still regenerate the erased concept from a fully noised seed and the original prompt when guided by this latent classifier (Table 4).

When combined with the Noise-Based Probe (Section 3.3), classifier guidance further amplifies recovery, yielding roughly a $1.5\times$ increase in classification accuracy for UCE, ESD-X, and ESD-U relative to standard classifier guidance alone.

The classifier-guided probe provides a powerful test of residual knowledge: rather than altering the text or context, it steers the model directly within its latent space. We note, however, that as discussed in Section 5, optimization-based probes such as this one may recreate concept-like outputs through the optimization process itself rather than strictly reveal residual representations.

| Metric | GA | UCE | ESD-x | ESD-u | TaskVec | STEREO | RECE |
|---|---|---|---|---|---|---|---|
| **Standard Classifier Guidance** (↓) | | | | | | | |
| CLIP | 26.3 | 28.2 | 28.1 | 28.1 | 27.6 | **25.7** | 27.1 |
| Class Acc. (%) | **3.7** | 45.6 | 47.8 | 46.7 | 30.2 | 5.8 | 33.3 |
| **Classifier-Guided Noise-Based Probing** (↓) | | | | | | | |
| CLIP | **26.5** | 29.1 | 28.6 | 28.4 | 27.8 | 27.1 | 27.2 |
| Class Acc. (%) | **4.1** | 75.6 | 73.3 | 59.1 | 35.1 | 20.3 | 36.7 |

Table 4: Classifier-guided probing results. Standard classifier guidance alone reveals residual concept signals across several erasure methods, while combining it with noise-based probing (*Classifier-Guided Noise Probe*) further amplifies the recovery of erased concepts. Notably, classification scores of recovered concepts for STEREO are more than double those achieved by Textual Inversion.

Prompt = "a picture of an airliner"

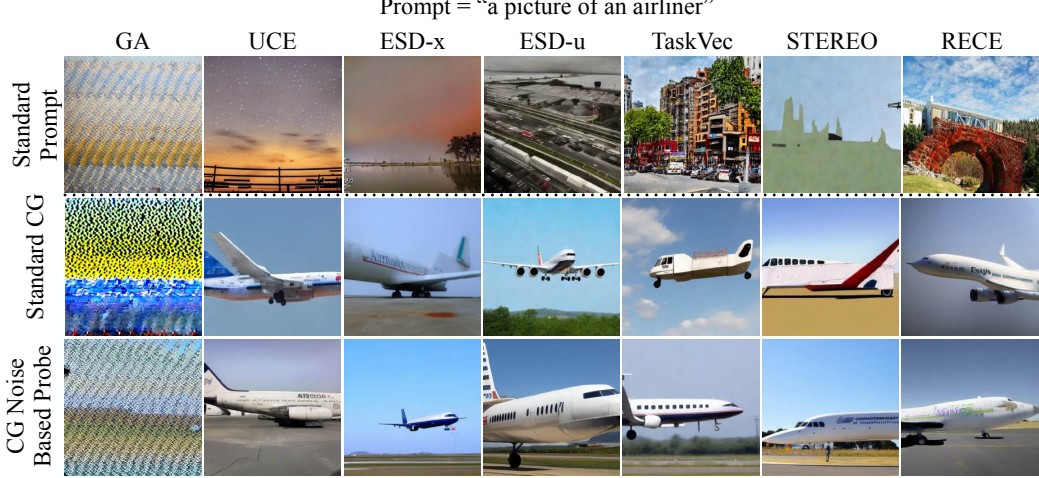

Figure 6: Classifier Guidance (CG) with Noise-Based probes significantly improves the visual fidelity and semantic accuracy of recovered airplane concepts. Generated airplanes exhibit more realistic structural features and better-defined geometries compared to the baseline method (particularly for Task Vectors and STEREO).

## 3.5 Dynamic Concept Tracing

**Question 5:** How does a concept evolve when progressively being erased?

We analyze the trajectories of the alternative generations for different erasure strengths. Namely, we prompt the model at different stages using the concept name in the prompt and inspect the resulting images. We find that methods which perform more robust erasure when evaluated using other probes tend to degrade the concept more consistently. That is, Gradient Ascent and Task Vector often converge to generate similar images along the trajectory, and these images degrade in quality as the erasure progresses. In contrast, methods that tend to behave in a way more similar to the *guidance-based* avoidance conceptual model exhibit more abrupt transitions between alternative generations. Namely, ESD-u and ESD-x produce more varied outputs when prompted with the erased concept (Fig. 7). To validate this observation, we measure the distance between CLIP embeddings of images generated by the original (unedited) model and those from the edited models (see App. D.6).

One potential way to understand this result is through the nature of the erasure mechanism. *Destruction-based* methods may create a 'dip' in the unconditional likelihood landscape by reduc-

ing the probability of generations containing the target concept (see right panel, Fig. 1). This may supress the generation probability not only for the target concept but also for semantically nearby concepts. Consequently, stronger erasure (deeper 'dips') may drive alternative generations further from the original concept. In contrast, *guidance-based* methods interfere with the guidance mechanism, but not necessarily with the unconditional probability. They may have less impact on the generated image structure while more significantly affecting which specific generation is selected.

We note that this possible explanation relies on assuming a given method is *guidance-* or *destruction-*based. Yet, the categorization of any specific method as one of our conceptual mechanisms remains tentative. While we believe concept tracing provides valuable complementary insight into erasure dynamics within our evaluation suite, drawing exact conclusions on the underlying mechanisms requires further study.

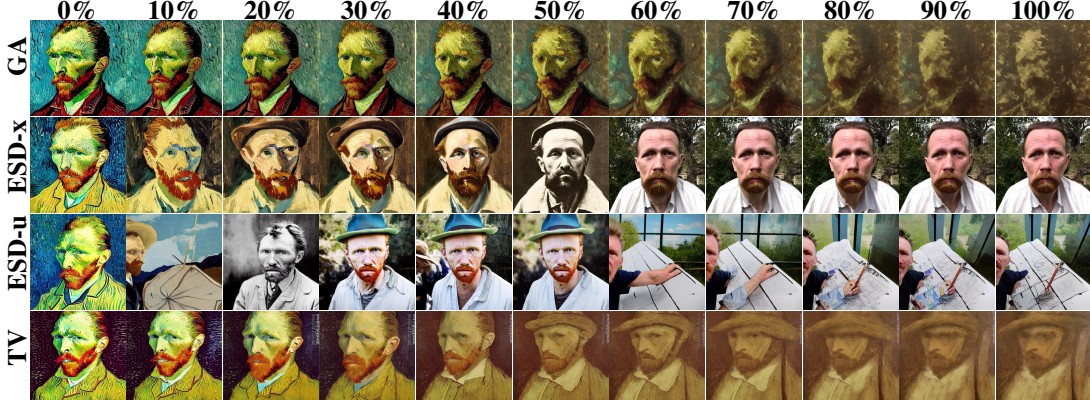

Figure 7: When comparing generations as concepts are progressively erased, the differences between method types become visually apparent. Methods that align more with the destruction-based conceptual model degrade concept generation continuously. In contrast, methods that align with the guidance-based conceptual model interfere with the conditional guidance process, producing more diverse images.

### 3.6 Summary

Across our four probes, distinct patterns emerged. Methods such as GA, TaskVectors, STEREO, and RECE, appear robust to text-based attacks like Textual Inversion and UnlearnDiffAtk (Sec. 3.1). Yet TaskVectors and RECE resurfaced the erased concept under inpainting and diffusion-completion probes (Sec. 3.2). Noise-based probing further showed that concepts erased by UCE, ESD-x, and ESD-u could be recovered with minor stochastic perturbations: suggesting that these models still encode the concept but weakly guide away from it (Sec. 3.3). Classifier-guided probing revealed even stronger residual traces: many models, including STEREO, could regenerate erased concepts at accuracies rivaling or exceeding Textual Inversion (Sec. 3.4). Finally, dynamic tracing exposed differing erasure dynamics, where GA and TaskVectors gradually degrade the concept, while ESD-x and ESD-u produce abrupt, unstable alternatives (Sec. 3.5).

Together, these findings indicate that most current erasure methods operate through guidance-based avoidance rather than true destruction of underlying representations, underscoring the need for multi-perspective evaluation to assess what "erasure" truly means in diffusion models.

## 4 Related Works

**Concept erasure methods for text-to-image models.** Recently, various techniques have been introduced to prevent generative models from producing images of unwanted concepts. Some work [2, 30, 22, 16] propose modifying the inference process to steer outputs away from unwanted concepts. Other methods utilize classifiers to adjust the generated results. However, since inference-guided approaches can be circumvented with sufficient access to model parameters [24], subsequent research has focused on directly updating the model weights. Pham et al. [19] apply task vectors

to shift the model towards a weight space that forgets the unwanted concepts. Heng and Soh [13] utilize continual learning techniques to erase targeted concepts. Gandikota et al. [8] fine-tune the model to minimize the likelihood of generating the desirable concepts. Gandikota et al. [10] propose a closed-form expression of the weights of an erased model. Gong et al. [11] used a closed-form solution to find target embeddings of a concept which are used to update the cross-attention layers accordingly. Zhang et al. [31] suggest cross-attention re-steering to update the cross-attention maps in the UNet model of Stable Diffusion to erase concepts.

**Attacks against concept erasure methods.** While concept erasure methods effectively prevent undesirable generations when the concept is explicitly mentioned in the prompt (e.g., "a painting in the style of Picasso"), recent studies have demonstrated that adversarial inputs can bypass most of these defenses. In a white-box setting, Pham et al. [18] leveraged textual inversion to learn word embeddings capable of reintroducing so-called erased concepts. Similarly, Rusanovsky et al. [21] applied the same technique to learn latent seeds that reconstruct the removed concepts. Other research [29, 32, 3] has focused on directly crafting hard prompts that evade concept erasure mechanisms.

**Internal representations in diffusion models.** Recent work has revealed that diffusion models encode semantic information in structured and interpretable ways. For instance, Gandikota et al. [9] demonstrated that semantic directions within the model can be effectively captured using low-rank adaptors, enabling precise continuous control. Building on this understanding, Dravid et al. [5] showed that semantic representations are localized within specific subspaces of the model's cross-attention weights. Further investigations into the architectural components of diffusion models have yielded important insights. Liu et al. [17], Surkov et al. [27] discovered that specific concepts can be modified by targeting sparse sets of neurons. Through the application of Sparse Autoencoders (SAEs). Toker et al. [28] leveraged the UNet as an analytical tool to probe text encoder representations by studying how different internal representations influence the final generated outputs. Through our holistic evaluation framework, we analyze how different erasure methods distinctly affect concept representations.

## 5   Limitations

**Causality and control in concept erasure.** In many cases, even the expectations for an *ideal* concept erasure algorithm remain unclear. For example, when attempting to erase an art style like Van Gogh's, should we also remove related styles, such as Edvard Munch's? This is particularly tricky when causality is involved (e.g., should erasing 'Van Gogh' cause the erasing of 'Edvard Munch' but not vice versa?). In any case, achieving this level of control is still beyond the capabilities of current methods. Nevertheless, our findings offer some guidance: destruction-based removal tends to impact related concepts more significantly than guidance-based avoidance.

**Evaluating other concepts.** Our study covers 13 concepts, 10 objects and 3 art styles. However, other concepts may include verbs, relationships, or abstract ideas (e.g., 'violence'). Studying such concepts is beyond the scope of this work.

**Discovery of existing knowledge vs. invoking it.** In optimization-based probing techniques, there is an inherent danger that the discovered knowledge does not originate from the original model, but rather to the optimization process [18]. Namely, it could be that an extensive enough optimization-based probe (such as shown in Sec.3.4) may be able to induce generation of a concept the model did not even encounter during training. Careful consideration of this possibility is required, and may depend on the exact purpose of the erasure (e.g., safety, intellectual property law, or privacy).

## 6   Conclusion

In this paper, we propose a suite of independent probing techniques to uncover traces of supposedly erased knowledge in diffusion models. Our evaluation reveals that knowledge undetectable through one technique can often be recovered through others. Surprisingly, probes that require less supervision sometimes prove more effective than their more complex counterparts. Our study is motivated by the suspicion that many existing methods merely redirect generation away from target concepts rather than thoroughly erasing them, whether explicitly or implicitly. We hope our categorization of guidance-based versus destruction-based mechanisms will guide future research into understanding how erasure fundamentally alters a model's internal representations and generative dynamics.

# Acknowledgements

RG and DB are supported by Open Philanthropy and NSF grant #2403304

NC was partially supported by the Israeli Data Science Scholarship for Outstanding Postdoctoral Fellows (VATAT).

We thank the anonymous NeurIPS reviewers for their valuable comments and suggestions that have substantially improved this manuscript.

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

## A    Broader Impact

Our work aims to improve the reliability and transparency of concept erasure in diffusion models, a task with growing importance as generative models are deployed in real-world settings. Effective erasure can help prevent the generation of harmful, private, or copyrighted content, but it also introduces risks, such as the potential misuse of these methods to suppress culturally or politically significant concepts. By providing a comprehensive evaluation framework, we offer tools to better understand the trade-offs involved in erasure methods, particularly between robustness and the preservation of unrelated capabilities. We hope this encourages more responsible use and assessment of erasure techniques, while recognizing that such methods are not a complete solution and must be applied with care and oversight.

## B    Training Free Inference Time Noise-Based Probe

Song et al. [25] introduced Denoising Diffusion Implicit Models (DDIM), which presented a deterministic generative process defined by the equation:

$$x_{t-1} = \sqrt{\alpha_{t-1}} \left( \frac{x_t - \sqrt{1 - \alpha_t} \epsilon_\theta^{(t)}(x_t)}{\sqrt{\alpha_t}} \right) + \underbrace{\sqrt{1 - \alpha_{t-1} - \sigma_t^2} \cdot \epsilon_\theta^{(t)}(x_t)}_{\text{“direction pointing to } x_t \text{”}} + \underbrace{\sigma_t \epsilon_t}_{\text{random noise}} \quad (5)$$

We observe that the random noise term acts as a brownian motion component, driving stochastic sample generation when $\sigma_t > 0$. This insight motivates our approach: by controlling the magnitude of this brownian motion, we can systematically explore a broader latent space of the diffusion model. We modify the DDIM formulation by introducing a scaling factor $\tau$ an additional random noise component:

$$x_{t-1} = \sqrt{\alpha_{t-1}} \left( \frac{x_t - \sqrt{1 - \alpha_t} \epsilon_\theta^{(t)}(x_t)}{\sqrt{\alpha_t}} \right) + \underbrace{\sqrt{|1 - \alpha_{t-1} - \eta \cdot \sigma_t^2|} \cdot \epsilon_\theta^{(t)(x_t)} + \eta \cdot \sigma_t \epsilon_t}_{\text{eta-based noise injection}} \quad (6)$$

We take the absolute value of the "direction pointing to $x_t$" term because scaling $\eta > 1$ can cause the square root to receive a negative argument, which breaks the generative process. To avoid this failure mode while still injecting increased stochasticity, we apply the absolute value inside the square root. This allows us to safely explore values of $\eta > 1$, enabling stronger noise injection than what standard DDIM or DDPM configurations permit. We leverage this controlled over-noising as a *training-free Noise-Based probe*, allowing the model to access otherwise unreachable latent regions that may contain residual concept information.

### B.1    Noising Probe in Practice

**Remark.**    The theoretical argument above suggests that injecting noise into the diffusion process can enable the recovery of erased concepts. Standard samplers such as DDPM already introduce stochasticity, but their noise levels are typically fixed and moderate. To evaluate the practical effect of our noising attack, we compare three sampling strategies: (1) `DDIM1`, the deterministic DDIM sampler with $\eta = 1$; (2) `DDPM`, the stochastic ancestral sampler; and (3) `Noise-Based`, our proposed sampling strategy that explicitly increases the noise level beyond standard settings (e.g., $\eta > 1$). DDIM with $\eta = 0$ is the standard deterministic scheduler for the models, which has been evaluated to produce the target concepts close to 0% of the time.

Table 5 reports CLIP similarity scores and top-1 classification accuracy across 13 erased concepts for three erasure methods: `esdu`, `esdx`, and `uce`. Our noising approach consistently improves both CLIP alignment and classification accuracy, demonstrating that inference-time noise injection can serve as a practical, training-free mechanism for concept recovery. See Section D.3 for details on noise scales and sampling configurations.

## C    Erasing and Evaluation Implementation Details

### C.1    Concepts

We consider a set of 10 object concepts: English Springer Spaniel, airliner, garbage truck, parachute, cassette player, chainsaw, tench, French horn, golf ball, and church; alongside 3 distinct art styles: Van Gogh, Picasso, and Andy Warhol. This selection allows us to evaluate the impact of concept erasure across both tangible objects and artistic styles, ensuring a diverse range of visual and semantic attributes in our analysis.

### C.2    Experiment Results Reproducibility

To train all the models, run the entire evaluation suite, and create the CLIP and classificatio metrics, we used two NVIDIA A6000 GPUs. This mainly involved generating the probingimages (for validating erasure, assessing

|  | esdu | esdx | uce |
|---|---|---|---|
| **CLIP Score** (↑) | | | |
| DDIM1 | 25.80 | 26.34 | 26.81 |
| DDPM | 25.80 | 26.35 | 26.81 |
| Noised-Based | **27.99** | **30.56** | **30.65** |
| **Top-1 Accuracy** (%) (↑) | | | |
| DDIM1 | 13.77 | 20.62 | 18.31 |
| DDPM | 13.00 | 20.38 | 16.85 |
| Noised-Based | **27.70** | **30.70** | **21.90** |

Table 5: Average CLIP scores and Top-1 classification accuracy for each method and sampling scheduler. The Noised-Based probe significantly boosts concept recovery performance.

robustness via attacks, and checking interference with unrelated concepts), and evaluating the performance using CLIP similarity and classification accuracy.

## C.3 Evaluation Protocol

### C.3.1 CLIP Evaluation

All similarity assessments were performed using CLIP ViT (`openai/clip-vit-base-patch32`). For an object, such as a garbage truck, we compared the output image to the generation prompt, i.e. "a picture of a garbage truck".

### C.3.2 Classifier Evaluation for Object Concepts

To assess whether erased concepts remain recognizable in generated images, we perform classification using a ResNet-50 model pretrained on the Imagenette dataset, a simplified subset of ImageNet. Each generated image is processed by the classifier, and the top-5 predicted class labels are extracted based on softmax scores.

We consider a prediction correct if the concept name (e.g., `cassette_player`) matches the top-1 prediction (Top-1 Accuracy), or appears anywhere in the top-5 predictions (Top-5 Accuracy). For each match, we also record the classifier's confidence score as the Top-1 or Top-5 Score.

Classification results are aggregated across all object concepts (excluding artistic styles), and we report the following metrics per method and evaluation setting:

- **Top-1 Accuracy**: Percentage of images where the correct label is the highest scoring prediction.
- **Top-5 Accuracy**: Percentage of images where the correct label appears within the top-5 predictions.
- **Top-1 / Top-5 Scores**: Average softmax score for correct labels when they appear in the top-1 or top-5.

This evaluation provides a quantitative measure of whether erased concepts can still be semantically identified using an external classifier trained on real-world object categories. For the main paper, we mainly focus on Top-1 Accuracy scores.

For artist classification, we used CLIP-based similarity scores between generated images and artist-specific prompts as a proxy for classification, following prior work on zero-shot image-text alignment.

### C.3.3 Side-effects on Unrelated Concepts

To evaluate whether concept erasure negatively impacts unrelated generations, we assess each model's ability to generate images for concepts that were not erased. Specifically, for each of the 13 erased concepts, we consider the remaining 12 as control classes. For each model, we generate 10 images per control class, resulting in 120 control task images. We then compute CLIP similarity scores and classification accuracies for these images to quantify the extent to which erasure methods affect generalization and performance on unrelated concepts. Please see Figure 8 for examples.

## C.4 Erasure Methods

To evaluate the impact of different concept erasure techniques, we implemented several existing methods and trained models under controlled settings. Below, we detail the exact configurations for each approach:

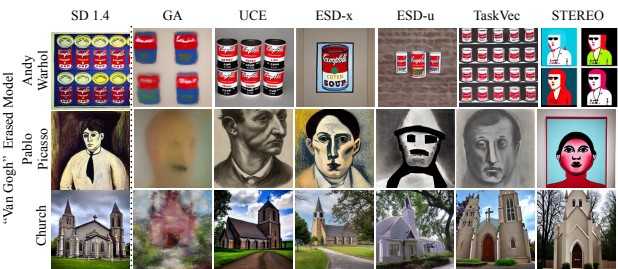

Figure 8: We show the undesirable effects of the erasure methods when erased "Van Gogh" concept on unerased concepts like "Church" and "Picasso". We find that Gradient Ascent and STEREO have the most interference with unrelated concept.

### C.4.1 Gradient Ascent (GA)

We implement the Gradient Ascent (GA) method by negating the standard training loss used in Stable Diffusion, effectively encouraging the model to increase the likelihood of generating the target concept. The training data comprises 500 diverse images per concept, generated using the original model along with their associated text prompts. For the *English Springer Spaniel* and *Garbage Truck* concepts, we reduced the number of fine-tuning steps to 10, while using 60 steps for all other concepts. To prevent degradation of the model's general utility, a known issue when applying GA over extended training, we adopt a conservative training configuration: a batch size of 5, gradient accumulation steps of 4, and a learning rate of $1 \times 10^{-5}$.

### C.4.2 Erased Stable Diffusion (ESD-x & ESD-u)

We fine-tuned for 200 steps using a learning rate of $2 \times 10^{-5}$.

### C.4.3 Unified Concept Editing (UCE)

We fine-tuned for 200 steps with an empty guiding concept and an erase scale of 1.

### C.4.4 Task Vector (TV)

To get the fine-tuned model for computing task vectors, we fine-tuned each model on 500 images for 200 steps, using a learning rate of $1 \times 10^{-5}$. We used batch size of 4 and gradient accumulation step of 4. For erasure, we set the editing strength $\alpha = 1.75$.

## D  Probing Methods and Additional Results

To assess the resilience of erasure methods against adversarial strategies, we conducted various attack experiments using a dataset of 100 prompts spanning 13 concepts (10 objects, 3 styles), each evaluated using unique seeds.

### D.1  Textual Inversion

Training involved 100 images, optimized for 3000 steps using a learning rate of $5 \times 10^{-4}$.

### D.2  UnlearnDiffAtk

The model was trained using a learning rate of 0.01 and a weight decay of 0.1, with the classifier parameter set to $K = 3$. ImageNet was used as the classifier for object-based erasures, while a custom classifier from the UnlearnDiffAtk repository was used for artist styles. Due to computational cost, UnlearnDiffAtk was evaluated on 30 prompts per concept, with 40 samples per experiment, where each sample was generated through 40 optimization steps.

### D.3  Inference-Time Noising Probe

We searched over an evenly spaced set of 6 $\eta$ values between 1.0 and 1.85: [1.0, 1.17, 1.34, 1.51, 1.68, 1.85]. These bounds were chosen based on qualitative observations: values above 1.85 produced overly noisy images, while those below 0.95 resulted in blurry object generation. For each $\eta$, we scaled the additional random

noise added at each denoising step by a factor of 1.00, 1.02, 1.03, or 1.04. A full grid search across these combinations yielded 24 samples per prompt per experiment. The CLIP model then selected the image with the highest similarity score as the representative probing instance.

## D.4 Inpainting

The inpainting pipeline was based on Stable Diffusion 1.5 and implemented via Hugging Face's `StableDiffusionInpaintPipeline`. Base images were $512 \times 512$ pixels and were masked with a $225 \times 225$ white box at the center. Source images were generated using Stable Diffusion 1.4. CLIP scores were computed only on the masked area to prevent artificially inflated similarity scores.

## D.5 Steered Latent Probing: Classifier Guidance Implementation

We implement steering latent probing by training a lightweight, timestep-aware classifier in the Stable Diffusion latent space and injecting its gradient during sampling to steer trajectories toward residual concept regions.

### D.5.1 Dataset Construction

We construct binary datasets per concept from ImageNet-1k as follows:

- **Positives:** all samples belonging to the target ImageNet class (resolved via the label name lookup in the HF metadata).
- **Negatives:** a diverse pool sampled from all other classes (default: $n_{\text{neg}} = 5000$), oversampled $5\times$ initially to ensure diversity, then downsampled to the requested size.
- **Binary label:** we add a `label_bin` column (1 for positives, 0 for negatives), and save each concept subset to disk as a HuggingFace Dataset.

We apply standard SD image preprocessing (resize to $512 \times 512$, normalization to $[-1, 1]$). The split is 90% train / 10% validation.

### D.5.2 Latent Encoding and Caching

Images are mapped to Stable Diffusion latents using the SD v1.4 VAE (`AutoencoderKL`), sampling from the posterior and scaling by $0.18215$, yielding latents of shape $(4, 64, 64)$. We **precompute and cache** latents for both train and validation splits to avoid repeated VAE passes during classifier training.

### D.5.3 Timestep-Aware Latent Classifier

Let $x \in \mathbb{R}^{4 \times 64 \times 64}$ denote a latent and $t \in \{0, \ldots, T-1\}$ a diffusion timestep. We encode $t$ using the scheduler's cumulative noise level $\bar{\alpha}_t$:

$$e(t) = [\bar{\alpha}_t, 1 - \bar{\alpha}_t] \in \mathbb{R}^2.$$

The classifier $f_\phi(x, t)$ is a small MLP with two streams:

- **Latent stream:** flatten $x$ and project to 1024 dims.
- **Timestep stream:** project $\mathbf{e}(t)$ to 1024 dims.

We sum the two 1024-d embeddings and pass through: SiLU, Dropout(0.3), Linear(1024→512), SiLU, Dropout(0.3), Linear(512→1), yielding a logit. This architecture is intentionally small to avoid overfitting and to keep gradients stable during guidance.

### D.5.4 Training Procedure

We train with AdamW (lr $1 \times 10^{-4}$, weight decay $10^{-3}$), batch size 8, gradient clipping at 1.0, for 10 epochs by default. The loss is BCEWithLogits with a positive-class weight

$$w_{\text{pos}} = \frac{\#\text{neg}}{\#\text{pos}}$$

computed from the subset to counter class imbalance.

To improve robustness across noise levels, each mini-batch is augmented with $k$ noisy views per sample (default $k = 7$). We draw timesteps via a power-law that favors noisier latents:

$$t \sim (\text{Uniform}(0, 1)^{1/\gamma}) \cdot (T - 1), \quad \gamma = 3.0,$$

and form noisy latents with the scheduler's forward operator. Validation averages logits over 3 independently sampled timesteps. After training for 70 epochs, we pick the checkpoint with the lowest validation loss.

### D.5.5 Inference-Time Steering (Guidance)

During sampling, we run the standard classifier-free guidance (CFG) pass to obtain $\epsilon_{\text{cfg}}$ and then *inject* the latent-classifier gradient. At each timestep $t$:

1. Compute $\epsilon_{\text{cfg}}$ using CFG with guidance scale 7.5.

2. Enable gradients on the current latent $x_t$ and compute the BCE loss with target label 1:

$$\ell_t = \text{BCEWithLogits}(f_\phi(x_t, t), 1).$$

3. Backpropagate to obtain $g_t = \nabla_{x_t} \ell_t$, scale by $s_{\text{clf}}$ (the *classifier guidance strength*), and rescale by the current noise level:

$$\tilde{\epsilon} = \epsilon_{\text{cfg}} - \sqrt{1 - \bar{\alpha}_t}\, s_{\text{clf}}\, g_t.$$

4. Step the scheduler with $\tilde{\epsilon}$ using a DDIM sampler then continue.

The classifier gradient acts as a steering force that nudges the diffusion trajectory toward regions where the classifier predicts high probability for the target concept. Since we're probing for *residual* concept knowledge in an erased model, successful steering reveals that the model still retains information about the supposedly erased concept.

This steering follows the principle of classifier guidance in diffusion models. To sample from a conditional distribution $p_\theta(\mathbf{x}_t \mid c^*)$, Bayes' rule tells us we can decompose the score as:

$$\nabla_{\mathbf{x}_t} \log p_\theta(\mathbf{x}_t \mid c^*) = \nabla_{\mathbf{x}_t} \log p_\theta(\mathbf{x}_t) + \nabla_{\mathbf{x}_t} \log p_\phi(c^* \mid \mathbf{x}_t)$$

The first term is the unconditional diffusion score, and the second term (approximated by our classifier gradient) steers the sampling toward the concept $c^*$. The guidance scale $s_{\text{clf}}$ controls how strongly we condition on the concept, effectively amplifying any residual concept knowledge that remains after erasure.

## D.6 Dynamic Concept Tracing

We quantitatively analyze how concept representations evolve during the erasure process by tracking the trajectories of generated images in CLIP embedding space. For each erasure method and target concept, we generate 25 images at varying erasure strengths and compute the centroid of their CLIP embeddings. This allows us to measure how far the generated concepts drift from their original representations as erasure intensity increases.

Figure 9 reveals distinct patterns between erasure methods. Gradient Ascent and Task Vector exhibit approximately linear trajectories, progressively pushing concept representations away from their original locations in embedding space. In contrast, ESD-x and ESD-u demonstrate more abrupt displacement. This pattern hints that their approach redirects generation toward unconditional outputs rather than fundamentally altering the concept representation itself.

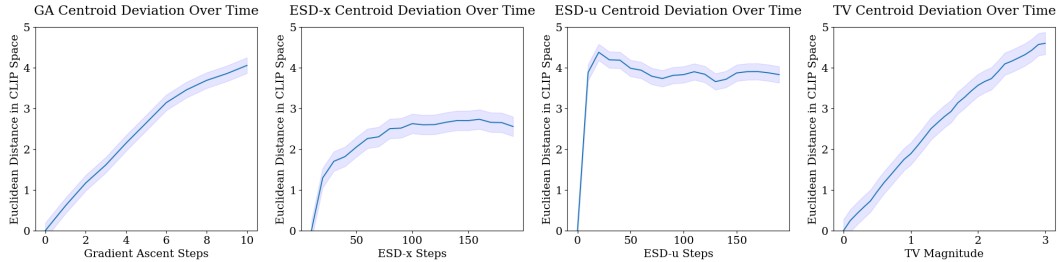

Figure 9: Trajectories of concept representations during erasure in CLIP embedding space. The x-axis shows erasure strength, while the y-axis measures the Euclidean distance between the centroid (mean) of all generated samples and the original concept embedding (computed from 25 samples per concept). Shaded regions indicate 95% confidence intervals. The GA and TV methods show more consistent degradation patterns than ESD-x and ESD-u.

# E Full Results with Standard Deviations

We include here the full quantitative results with standard deviations across runs, complementing Tables 1, 2, and 3 from the main paper. These tables report the mean and standard deviation for both CLIP similarity scores and classification accuracies across multiple erasure evaluation settings. Including standard deviation helps illustrate the consistency and robustness of each erasure method under different probing strategies.

| Eval Metric | Base | GA | UCE | ESD-x | ESD-u | TaskVec | STEREO | RECE |
|---|---|---|---|---|---|---|---|---|
| **Erased Concepts** ($\downarrow$) | | | | | | | | |
| CLIP | – | 24.3 ± 2.7 | 22.4 ± 5.2 | 21.1 ± 4.1 | 20.9 ± 3.4 | 23.1 ± 3.0 | **19.6 ± 2.3** | 21.2 ± 4.0 |
| **Text Inversion** ($\downarrow$) | | | | | | | | |
| CLIP | – | **22.7 ± 2.5** | 30.7 ± 2.0 | 30.6 ± 2.4 | 28.0 ± 3.4 | 25.1 ± 2.6 | 24.5 ± 2.9 | 29.2 ± 2.8 |
| **UnlearnDiffAtk** | | | | | | | | |
| CLIP | – | **26.0 ± 2.2** | 28.3 ± 3.2 | 28.7 ± 2.2 | 27.8 ± 2.8 | 27.1 ± 1.7 | 26.1 ± 2.8 | 27.9 ± 2.3 |
| **Unrelated Concepts** ($\uparrow$) | | | | | | | | |
| CLIP | – | 28.8 ± 2.8 | **31.2 ± 2.3** | 30.8 ± 2.5 | 30.7 ± 3.3 | 29.4 ± 2.6 | 29.0 ± 3.0 | 30.5 ± 2.7 |
| **Inpainting** ($\downarrow$) | | | | | | | | |
| CLIP | 29.5 ± 2.2 | 24.8 ± 2.4 | 26.9 ± 3.3 | 26.8 ± 3.1 | 23.9 ± 3.1 | 25.9 ± 2.6 | **22.7 ± 3.0** | 26.3 ± 2.7 |
| **Diffusion Completion** $t = 5$ ($\downarrow$) | | | | | | | | |
| CLIP | 30.2 ± 2.1 | 24.0 ± 2.4 | 27.7 ± 2.8 | 27.2 ± 3.1 | 26.9 ± 2.9 | **23.8 ± 2.4** | 23.9 ± 2.7 | 28.8 ± 2.5 |
| **Diffusion Completion** $t = 10$ ($\downarrow$) | | | | | | | | |
| CLIP | 30.2 ± 2.1 | **24.5 ± 2.3** | 29.6 ± 2.3 | 28.7 ± 2.9 | 27.5 ± 2.8 | 24.9 ± 2.3 | 27.8 ± 2.6 | 28.8 ± 2.5 |

Table 6: CLIP scores (mean ± std) across concept erasure methods. Lower scores ($\downarrow$) indicate better erasure of the target concept, while higher scores ($\uparrow$) reflect stronger retention of unrelated concepts. Rows cover adversarial and in-context evaluations including inpainting and diffusion completion at denoising steps $t = 5$ and $t = 10$.

| Eval Metric | Base | GA | UCE | ESD-x | ESD-u | TaskVec | STEREO | RECE |
|---|---|---|---|---|---|---|---|---|
| **Erased Concepts** ($\downarrow$) | | | | | | | | |
| Acc. (%) | – | 0.6 ± 0.48 | 4.4 ± 1.1 | 3.6 ± 1.3 | 1.0 ± 0.69 | 2.2 ± 1.0 | **0.0 ± 0.00** | 4.0 ± 1.2 |
| **Text Inversion** ($\downarrow$) | | | | | | | | |
| Acc. (%) | – | **0.6 ± 0.59** | 71.2 ± 2.3 | 65.9 ± 2.9 | 31.8 ± 3.6 | 6.2 ± 1.8 | 6.3 ± 1.6 | 58.2 ± 3.1 |
| **UnlearnDiffAtk** | | | | | | | | |
| Acc. (%) | – | 6.5 ± 1.5 | 26.8 ± 2.8 | 21.0 ± 2.6 | 16.6 ± 2.3 | 10.3 ± 2.1 | **3.7 ± 1.0** | 7.2 ± 1.7 |
| **Unrelated Concepts** ($\uparrow$) | | | | | | | | |
| Acc. (%) | – | 52.2 ± 2.7 | **75.0 ± 1.9** | 71.3 ± 2.2 | 70.4 ± 2.4 | 60.4 ± 2.6 | 52.8 ± 2.9 | 71.7 ± 2.1 |
| **Inpainting** ($\downarrow$) | | | | | | | | |
| Acc. (%) | 77.7 ± 1.5 | **61.7 ± 2.4** | 69.1 ± 1.8 | 69.1 ± 1.9 | 68.5 ± 1.7 | 66.8 ± 1.6 | 63.8 ± 2.0 | 68.2 ± 1.8 |
| **Diffusion Completion** $t = 5$ ($\downarrow$) | | | | | | | | |
| Acc. (%) | 78.0 ± 1.4 | **1.1 ± 0.58** | 42.7 ± 2.7 | 37.8 ± 3.0 | 32.5 ± 3.2 | 2.4 ± 0.94 | 3.2 ± 1.2 | 36.5 ± 3.3 |
| **Diffusion Completion** $t = 10$ ($\downarrow$) | | | | | | | | |
| Acc. (%) | 78.0 ± 1.4 | **3.2 ± 1.1** | 62.1 ± 2.3 | 54.8 ± 2.7 | 36.9 ± 3.0 | 6.1 ± 1.5 | 21.2 ± 2.2 | 45.4 ± 2.8 |

Table 7: Classification accuracy (%, mean ± std) across seven concept erasure methods and the original Stable Diffusion model (**Base**). Lower values ($\downarrow$) on erased concepts, textual inversion, UnlearnDiffAtk, inpainting, and diffusion completion indicate more effective removal of the target concept. Higher values ($\uparrow$) on unrelated concepts reflect successful preservation of general generation capabilities.

