# OpenReview forum: "When Are Concepts Erased From Diffusion Models?"
_NeurIPS.cc/2025/Conference — NeurIPS 2025 poster_

### Official Review · Reviewer_H6xi · 2025-06-27

**Clarity:** 3
**Significance:** 1
**Originality:** 1
**Rating:** 2
**Confidence:** 4

**Summary:**

This paper studies the robustness of concept removal in text2img diffusion models. The authors first propose a concept map of how the erasure methods work and then evaluate the methods with adversarial attacks, inpanting, and random pertubations.

**Questions:**

Please see weakness.

**Ethical Concerns:**

["NO or VERY MINOR ethics concerns only"]

**Final Justification:**

**Recommendation: Reject**

I recommend rejecting this paper due to significant concerns regarding its novelty, depth of analysis, and engagement with prior work. While the topic is relevant, the contribution is not substantial enough for publication at NeurIPS.

---

### **Key Weaknesses**

1.  **Incremental Contribution:** The paper's core contribution is applying existing attack methods to existing unlearning defenses. This is a straightforward combination of prior work that yields predictable results. The findings, while confirming that defenses are not perfectly robust, lack the novelty and surprise expected of a top-tier publication.

2.  **Superficial Analysis:** The work functions more as an experimental report than a deep scientific investigation. It demonstrates vulnerabilities but fails to provide a rigorous analysis of *why* these defenses fail. The absence of new theoretical insights, a novel framework, or a deeper understanding of the failure mechanisms limits its scientific impact.

3.  **Incomplete Literature Review:** The related work section is critically incomplete. It fails to cite or discuss several recent and relevant papers that have already established the unreliability of diffusion unlearning methods, including key works like [1], [2], and [3]. This omission undermines the paper's claims to originality and its overall scholarly foundation.

---

Lastly, it is widely recognized in the adversarial ML community that constructing robust defenses is significantly more challenging than devising attacks. An attack paper at a premier venue should therefore offer more than a simple demonstration of vulnerability; it should provide novel insights or uncover fundamental flaws. Accepting this work, which is primarily an application of existing techniques, would risk lowering the standard for publication and could inadvertently discourage the more difficult and necessary research into developing robust defenses.

**Limitations:**

Yes

**Paper Formatting Concerns:**

NaN

**Quality:**

2

**Strengths And Weaknesses:**

== Strength
1. The proposed concept map makes sense intuitively.
2. The paper covers several erasure methods and three attack perspectives.

== Weakness
1. The trajectory probing method introduced by the authors does not cover vulnerabilities beyond gradient-based and inpainting-based attacks, and the method itself is too naive. Since the ultimate goal is to uncover the supposedly erased concepts, why not using classifiers/conditions to guide the generation/searching process, which will be much stronger than random guessing.
2. There exists tension between erasure effect and side effect is well-known. This paper does not expand our knowledge in this trade-off with experiements/thoerys.
3. Validness of the evaluation metrices is not guranteed. Are the classifier/CLIP highly correlated with human perception? Should we introduce non-neural network based evaluations? Does it make sense to use an ensemble of classifiers to improve accuracy?
4. The concept map is not rigorously verified with experiments/theorys.

In summary, this paper is limited in contribution due to lack of technical novelty and new experiemental findings.

---

> ### Author Rebuttal · Authors · 2025-07-29
>
> We thank the reviewer for their feedback. We are glad you found our conceptual framework intuitive. We address specific concerns below:
>
> **"trajectory probing method … does not cover vulnerabilities beyond gradient-based and inpainting-based attacks” "using classifiers/conditions to guide the generation/searching process"**
>
> We appreciate this thoughtful suggestion. One of the key strengths of the noising probe is its simplicity and training-free nature, and it can outperform other evaluation methods in specific cases (see Figure 7 for example, or the NSFW study in response to Reviewer jc67).
>
> That said, your comment inspired us to explore classifier-guided generation as a complementary approach.  Following the method outlined in “Diffusion Models Beat GANs on Image Synthesis” (Dhariwal & Nichol), we implemented classifier guidance during generation. In addition to using their pre-trained classifier, we also trained a custom binary classifier for garbage truck on noised latents from Stable Diffusion 1.4 using the ImageNet-1k dataset.
>
> While classifier guidance alone provided some benefit, combining it with the noise-based probe significantly boosted performance: even surpassing UDA on the garbage truck class.
>
> |**Metric / Method**|ESDx|ESD-U|UCE|STEREO|
> |---|---|---|---|---|
> |OpenAI Classifier (CLIP, Acc %)|24.5, 2.0%|24.0, 1.5%|21.3, 3.2%|20.3, 0.0%|
> |SD 1.4 Classifier (CLIP, Acc %)|27.1, 5.0%|26.5, 4.1%|22.3, 6.0%|21.4, 0.5%|
> |Noise-Based Probe (CLIP, Acc %)|26.1, 5.0%|28.3, 4.4%|28.3, 4.4%|25.7, 1.0%|
> |Noise-Based Probe + OpenAI Classifier (CLIP, Acc %)|27.0, 7.0%|28.3, 28.4%|28.3, 28.4%|26.3, 1.3%|
> |Noise-Based Probe + SD 1.4 Classifier (CLIP, Acc %)|**29.3, 43.0%**|**28.3, 28.4%**|**29.4, 50.0%**|**26.7, 1.5%**|
>
> **“tension between erasure effect and side effect is well-known”**
>
> While we acknowledge that such tradeoffs have been observed, in our paper we further investigate this tension. To emphasize our findings, we report each method’s performance along three key axes: erasure effectiveness, FID score, and unrelated concept preservation.
>
> We define a Combined Method Score, which is an average of the method’s normalized erasing performance (across the different probes), unrelated concept preservation, and FID score. Our score reveals several interesting trends: STEREO outperforms ESD-U, the method it builds upon; and RECE similarly improves on its base method, UCE. Although both GA and STEREO achieve near-perfect erasure scores, their overall performance diverges: STEREO ranks highest overall, while GA ranks lowest.
>
>
> |**Rank**|**Method**|**Erasure**|**FID (↓)**|**Preservation**|**Model Score**|
> |---|---|---|---|---|---|
> |1|Stereo|**0.99**|19.5|0.74|**0.81**|
> |2|ESD-U|0.76|17.4|0.86|0.80|
> |3|TaskVec|0.90|18.8|0.79|0.80|
> |4|RECE|0.69|17.5|0.87|0.78|
> |5|UCE|0.60|**16.5**|**0.90**|0.76|
> |6|ESD-X|0.63|17.1|0.87|0.75|
> |7|GA|0.97|21.2|0.74|0.73|
> |8|Base (SD 1.4)|0.00|16.0|1.00|0.50|
>
>   **“Are the classifier/CLIP highly correlated with human perception?” “Should we introduce non-neural network based evaluations”**
>
> Thank you for the questions. Yes, classifiers are correlated with human perception and are often used for similar purposes [1,2,3,4]. Furthermore, to complement our automated evaluation metrics (e.g., CLIP and classifier accuracy), we conducted a small-scale human evaluation to assess whether erased concepts were still recognizable to non-expert observers.
>
>   We recruited human subjects to manually classify generated images for ESDX, UCE, and GA, and **3 concepts** (English Springer Spaniel, Cassette Player, Van Gogh) across **5 evaluation settings**:
> - Standard Prompt
>
> - Inpainting
>
> - Noise-Based Probe
>
> - Textual Inversion
>
> - UnlearnDiffAtk
>
> Each setting included 20 images per concept, for a total of **900 images** (3 models x 3 concepts × 5 evaluations × 20 images).
>
> Below, we report the average human classification accuracies (in %) alongside the automated classifier results from the main paper for comparison:
>
> |**Evaluation**|**Automated Classifier Accuracy (%)**|**Human Classification Accuracy (%)**|
> |---|---|---|
> |Standard Prompt|2.3|3.2 ± 0.9|
> |Inpainting|61.7|64.3 ± 3.2|
> |Noise-Based Probe|17.9|16.1 ± 1.3|
> |Textual Inversion|45.2|41.7 ± 2.0|
> |UnlearnDiffAtk|18.3|18.5 ± 1.7|
>
>
>
> The human evaluation results generally aligned with our automated metrics. Across all five evaluation settings and four models, the relative performance ordering was preserved: GA produced the lowest recognition rates (e.g., 1.3% human vs. 0.6% automated on standard prompts), while human classification of inpainting results for ESD-x and GA averaged 67.5% and 75.6%, closely tracking automated results in 69.1% for ESD-X and 61.7% for GA.
>
>
>
> These observations suggest that while human and automated judgments mostly agree, human evaluation provides complementary insights, particularly in borderline cases where visual quality or conceptual ambiguity may affect recognition.
>
>
>
> [1] Zhang, Richard, et al. "The unreasonable effectiveness of deep features as a perceptual metric." Proceedings of the IEEE conference on computer vision and pattern recognition. 2018.
>
> [2] Kumar, Manoj, et al. "Do better ImageNet classifiers assess perceptual similarity better?." Transactions on Machine Learning Research.
>
> [3] Gandikota, Rohit, et al. "Unified concept editing in diffusion models." Proceedings of the IEEE/CVF Winter Conference on Applications of Computer Vision. 2024.
>
> [4] Gong, Chao, et al. "Reliable and efficient concept erasure of text-to-image diffusion models." European Conference on Computer Vision. Cham: Springer Nature Switzerland, 2024.
>
>
> **“Does it make sense to use an ensemble of classifiers to improve accuracy?”**
>
>  While we believe different classifiers tell a similar story, we are happy to report results with other classifiers as well. For example, see the table below, where we calculated the pearson correlation between other classifiers trained on ImageNet and the resnet-50 classifier we used:
>
>   |**Model**|**Overall Pearson r with ResNet-50**|
> |---|---|
> |EfficientNet-B4|0.796|
> |**ConvNeXt-Base**|**0.814**|
> |ViT-B/16|0.778|
>
> We find overall pearson correlations of ~0.8, which suggest strong alignment between the classifiers.
>
> **“The concept map is not rigorously verified”**
>
> The concept map in Fig. 1 is intended to illustrate the two extremes of concept erasure we consider: guidance-based avoidance and destruction-based erasure. While the probing techniques we propose offer empirical insights into the erasing mechanisms of different methods, we acknowledge that a more rigorous theoretical foundation is needed. We view this work as a first step and hope it inspires further research from the community.
>
> Our motivation behind guidance vs destruction categorization is based on the resultant distribution of the erased concept post erasure. With guidance based methods, the unlearning algorithm pushes the conditional probability $P(X | c)$, where $c$ is the erased concept, to another known distribution. For example, with UCE we can write the erased distribution $P(X | c*)$ where $c*$ is the anchor concept, which comes from the following loss function:
>
> $$
> \mathcal{L}(W) = \sum_{c_i \in E} \left\| W c_i - v_i^* \right\|^2 + \sum_{c_j \in P} \left\| W c_j - W_{\text{old}} c_j \right\|^2
> $$
>
> Specifically, UCE learns a new attention projection matrix $\( W \)$ that maps erased concept prompts $\( c_i \)$ to substitute vectors $\( v^*_i \)$, while preserving the responses to protected concepts $\( c_j \)$.
>
> On the other hand, other methods (e.g., Gradient Ascent, TV) use alternative objectives such as increasing the diffusion loss. Even if we suspect that the erasing mechanism removes explicit knowledge of the target concept from the model, the resultant concept distribution is difficult to explicitly compute and categorize. To tackle this challenge, we have proposed our suite of evaluations described in the paper as an empirical tool for analyzing erased models.
>
> Thank you very much for your comments. If our clarifications address your concerns, we would greatly appreciate an updated assessment. If not, please let us know what can be further improved; we are happy to continue the discussion any time until the end of the discussion period. Thank you!

---

> > ### Comment · Reviewer_H6xi · 2025-08-04
> >
> > Dear authors:
> >
> > Thank you for the additional experiements and explanations, and I find the classifier-guidance experiment interesting. However, my core concern is still not addressed.
> >
> > To summarize my review, this paper seems to be a relatively comprehensive survey of existing unlearning methods for diffusion models, covering several exisitng unlearning methods and related metrics. And the core novel method is simply a random search algorithm.
> >
> > To be specific, it is below the acceptance bar of Neurips for a survey paper without (1) informative experienmental/theoretical insights. I cannnot draw more insights from the paper in its current state beyond "Method A is better Method B in some scenario". There is no in-depth experienmental study and no rigorous theoretical understanding of the reasons behind the phenomenon. (2) novel research perspective/framework that reshapes the community's understanding of the field like [1].
> >
> >
> > [1] Obfuscated Gradients Give a False Sense of Security: Circumventing Defenses to Adversarial Examples; ICML 2018.

---

> ### Author Response · Authors · 2025-08-04
>
> Thank you for giving us the opportunity to clarify about the main point of our paper:
>
> Our work is not a survey paper, rather we propose to robustly evaluate erasure method from a multiple-perspective lens. Most of the erasure methods in diffusion models, evaluate their erasure robustness through a single or narrow lens [1,2,3].  We found that **every** erasure method can be circumvented with  one or more of our suggested evaluations (e.g. in-context probing, gradient-based probing, diffusion trajectory, etc.). We believe our work will shift the current norm of evaluating robustness, from prompting based evaluation or textual inversion attacks, to a more holistic approach.
>
> [1] Schwinn, Leo, et al. "Soft prompt threats: Attacking safety alignment and unlearning in open-source llms through the embedding space." Advances in Neural Information Processing Systems 37 (2024): 9086-9116.
>
> [2] Pham, Minh, et al. "Circumventing Concept Erasure Methods For Text-To-Image Generative Models." The Twelfth International Conference on Learning Representations.
>
> [3] Rusanovsky, Matan, et al. "Memories of forgotten concepts." Proceedings of the Computer Vision and Pattern Recognition Conference. 2025.

---

### Official Review · Reviewer_Bk4k · 2025-06-29

**Clarity:** 3
**Significance:** 2
**Originality:** 2
**Rating:** 4
**Confidence:** 2

**Summary:**

This paper studies how existing approaches erase concepts from diffusion models and discusses their limitations. Two conceptual models, i.e., guidance-based avoidance and destruction-based removal, for the erasure mechanism in diffusion models are proposed. Furthermore, some probing methods are applied to evaluate existing removal methods.

**Questions:**

See above.

**Ethical Concerns:**

["NO or VERY MINOR ethics concerns only"]

**Final Justification:**

After the discussion, I still feel this paper could benefit more from deeper theoretical modeling and more empirical evidence of the difference between guidance-based avoidance and destruction-based removal. Moreover, as also supported by Reviewer H6xi, this paper does not provide a "novel research perspective/framework," which restricts the contribution and novelty. Nonetheless, I personally think this paper is still interesting, and the reasons to accept slightly outweigh reasons to reject. Therefore, I decide to maintain my current rating.

**Limitations:**

Yes.

**Quality:**

3

**Strengths And Weaknesses:**

Strengths
- The motivation is clear, and the problem of concept removal is fundamental.
- The paper is well-written.
- The proposed probing methods clearly present the difference among methods. These methods provide an effective framework for comprehensive evaluation of concept removal methods in the future.

Weaknesses
- Even though the two conceptual models are intuitive and easy to understand, this paper could benefit from deeper analysis and theoretical modeling of them. For example, what is the essential difference between guidance-based avoidance and destruction-based removal? If the target concept is always avoided in generation, can we also say it is a kind of reduction in its likelihood?
- This paper stops by identifying limitations of existing methods and fails to propose a better data erasure method. Therefore, I think the novelty is limited to some extent.

---

> ### Author Rebuttal · Authors · 2025-07-29
>
> We thank the reviewer for their thoughtful summary and positive feedback. We appreciate that the reviewer found the motivation clear and the problem well framed, and the probing methods effective. We are especially encouraged by your comment that the evaluation framework may serve as a useful tool for future work in this area.
>
> **“what is the essential difference between guidance-based avoidance and destruction-based removal?”**
>
> To clarify, we suggest that the key difference between guidance-based avoidance and destruction-based removal lies in the resulting distribution of the erased concept after erasure. With guidance based methods, the unlearning algorithm pushes the conditional probability $P(X | c)$, where $c$ is the erased concept, to another known distribution. For example, with UCE we can write the erased distribution $P(X | c*)$ where $c*$ is the anchor concept, which comes from the following loss function:
>
> $$
> \mathcal{L}(W) = \sum_{c_i \in E} \left\| W c_i - v_i^* \right\|^2 + \sum_{c_j \in P} \left\| W c_j - W_{\text{old}} c_j \right\|^2
> $$
>
> Specifically, UCE learns a new attention projection matrix \\( W \\) that maps erased concept prompts \\( c_i \\) to substitute vectors \\( v^*_i \\), while preserving the responses to protected concepts \\( c_j \\).
>
> On the other hand, destruction based methods fundamentally remove the knowledge of the concept, reducing the overall likelihood that samples from $P(X)$ exhibit the erased concept. A canonical example of this would be retraining the model on data that excludes the target concept. However, even if we suspect that the erasing mechanism removes explicit knowledge of the target concept from the model, the resultant concept distribution is difficult to explicitly compute and categorize. To tackle this challenge, we propose our suite of evaluations described in the paper as an empirical tool for analyzing these erased models. We will update the final draft to make this more clear.
>
> **"This paper stops by identifying limitations of existing methods"**
>
> While our paper focuses on identifying the limitations of existing erasure methods rather than proposing a new one, similar studies have often paved the way for developing stronger approaches [1,2,3,4]. For instance, the STEREO method (the most robust and highest ranked model in our evaluation framework) was developed in response to Pham et al.'s [1] finding that textual inversion can circumvent erasure in ESD. We hope our probing insights inspire more robust erasure strategies, such as integrating context-aware probes during training or leveraging the noise-based probe scheduling setup to better capture diverse generation paths.
>
>
>
> [1] Pham, Minh, et al. "Circumventing Concept Erasure Methods For Text-To-Image Generative Models." The Twelfth International Conference on Learning Representations.‏
> [2] Srivatsan, Koushik, et al. "Stereo: A two-stage framework for adversarially robust concept erasing from text-to-image diffusion models." Proceedings of the Computer Vision and Pattern Recognition Conference. 2025.‏
> [3] Robey, Alexander, et al. "Smoothllm: Defending large language models against jailbreaking attacks." arXiv preprint arXiv:2310.03684 (2023).
> [4] George, Naveen, et al. "The Illusion of Unlearning: The Unstable Nature of Machine Unlearning in Text-to-Image Diffusion Models." Proceedings of the Computer Vision and Pattern Recognition Conference. 2025.
>
> Thank you very much for your comments. We will include all the revisions in the final version of the paper. We are happy to continue the discussion any time until the end of the discussion period. Thank you!

---

> > ### Comment · Reviewer_Bk4k · 2025-08-04
> >
> > Thank you for your response! I still feel confused on the essential difference between guidance-based avoidance and destruction-based removal.
> >
> > - I feel there is no formal definition of the "resulting distribution." Intuitively, the generated results can be regarded as samples from the resulting distribution, which in turn defines the resulting distribution. Therefore, both so-called guidance-based avoidance and destruction-based removal reduce the probability of the erased concept, so I do not understand their essential difference, as they result in the same generations. Therefore, I feel this paper could benefit from a deeper theoretical modeling of the concept removal methods.
> > - By the way, I do not fully understand the provided loss function. The notations are not defined, and I do not understand their relationship with the erased distribution.

---

> ### Author Response · Authors · 2025-08-04
>
> Thank you for engaging in the discussion! We appreciate the further enquiry on "resultant distribution" and we would like to further clarify our earlier response:
>
> The "resulting distribution" can in some cases, be in-theory estimated based on the loss function of the erasure method. For example, most of the erasure methods aims to change the conditional distribution of the diffusion model \$P(\mathbf{X}_t|c)\$ where \$\mathbf{X}_t\$ is the noisy image and $c$ is the prompt for the erased concept (e.g. "Van Gogh").
>
>  - In the case of ESD [1], the method aims to change the conditional distribution to \$\frac{P(\mathbf{X}_t|c)}{(P(c/\mathbf{X}_t))^\eta}\$ (please refer to equation 4 in their paper [1]).
>
> - In the case of UCE [2], the method aims to change the conditional distribution to  \$P(\mathbf{X}_t|c*)\$ where $c*$ is the anchor concept prompt (e.g. "art")
>
> However, in other cases the resulting distribution is hard to determine. For example, Gradient Ascent aims to increase the diffusion loss under the conditional constraints (i.e. increase the diffusion loss when the model is prompted with the erased concept prompt). This makes it difficult to estimate the "resultant distribution". For this case, as the reviewer correctly pointed out, the way to estimate the distribution is post-hoc, by generating images using the erased model. Yet, generating images without any conditioning is unlikely to produce the target concept, even with the original model. As a result, verifying whether the concept has been erased is inherently challenging. This observation motivates the suite of evaluations presented in this paper.
>
> We hope this provides better clarity.
>
> **Regarding the paper's motivation:** We would also like to provide some more clarity on our main motivation. Most of the erasure methods in diffusion models, evaluate their erasure robustness through a single or narrow lens [3,4,5].  We found that **every** erasure method can be circumvented with either or multiple of our suggested evaluations (e.g. in context lens, gradient based lens, diffusion trajectory, etc.). We believe our work would change the current norm of evaluating robustness by simple prompting or textual inversion attacks, but rather take a holistic approach.
>
> [1] Gandikota, R., Materzynska, J., Fiotto-Kaufman, J., & Bau, D. (2023). Erasing concepts from diffusion models. In Proceedings of the IEEE/CVF international conference on computer vision (pp. 2426-2436).
>
> [2] Gandikota, R., Orgad, H., Belinkov, Y., Materzyńska, J., & Bau, D. (2024). Unified concept editing in diffusion models. In Proceedings of the IEEE/CVF Winter Conference on Applications of Computer Vision (pp. 5111-5120).
>
> [3] Schwinn, Leo, et al. "Soft prompt threats: Attacking safety alignment and unlearning in open-source llms through the embedding space." Advances in Neural Information Processing Systems 37 (2024): 9086-9116.
>
> [4] Pham, Minh, et al. "Circumventing Concept Erasure Methods For Text-To-Image Generative Models." The Twelfth International Conference on Learning Representations.
>
> [5] Rusanovsky, Matan, et al. "Memories of forgotten concepts." Proceedings of the Computer Vision and Pattern Recognition Conference. 2025.

---

> ### Author Response · Authors · 2025-08-04
>
> We would also like to clarify the notation of the UCE loss function:
> $$
> \mathcal{L}(W) = \sum_{c_i \in E} \left\| W c_i - v_i^* \right\|^2 + \sum_{c_j \in P} \left\| W c_j - W_{\text{old}} c_j \right\|^2
> $$
> In this loss:
> - $c_i \in E$ a token embedding in the set of erased concepts.
> - $c_j \in P$ a token embedding in the set of protected (non-erased) concepts.
> - $W$ is the new projection matrix of the cross attention projection layers learned during erasure.
> - $W_{\text{old}}$ is the original (pre-erasure) projection matrix.
> - $v_i^* $ is an "anchor" or substitute vector representing the target distribution for the erased concept $( c_i )$.
>
> The first term encourages the projection of the erased concept $( W c_i )$ to match the substitute vector $( v_i^* )$, pushing the generation with the erased concept toward a substitute target distribution. The second term ensures that projections of protected concepts remain close to their original representations, preserving unrelated knowledge.
>
> We present this example as a special case where erasure is expected to be guidance-based rather than destruction-based. That is, the diffusion model may still assign high likelihood to the erased concept, even if it fails to generate it from simple prompts, as demonstrated in the paper. However, many robust erasure methods rely on objectives that make it difficult to determine whether the concept is truly removed or merely avoided in specific contexts. Our empirical evaluation aims to distinguish between these two scenarios.

---

> > ### Comment · Reviewer_Bk4k · 2025-08-04
> >
> > Thanks for the detailed response, and I appreciate the time and efforts you have invested in the discussion. As mentioned above, I understand the main motivation.
> >
> > In my current understanding, the key difference between guidance-based avoidance and destruction-based removal is whether the probability of the erased concept gets reduced under unconditional generation, i.e., in the unconditional probability $P(X)$. I am not sure whether my understanding is correct, and I still feel this paper could benefit more from deeper theoretical modeling and more empirical evidence from this view.
> >
> > Moreover, as also supported by Reviewer H6xi, this paper does not provide a "novel research perspective/framework." Nonetheless, I think this paper is still interesting, and the reasons to accept outweigh reasons to reject. Therefore, I decide to maintain my current rating.

---

> > > ### Author Response · Authors · 2025-08-06
> > >
> > > We would like to thank the reviewer again for their thoughtful discussion.
> > >
> > > Yes, your current understanding is correct. In practice, we find that different practical methods achieve only approximate versions of these ideal cases of “pure” guidance- or destruction- based. We will clarify it better in the final version of the paper.
> > >
> > > Thank you very much, once again, for your excellent comments.

---

### Official Review · Reviewer_r56k · 2025-07-03

**Clarity:** 3
**Significance:** 2
**Originality:** 2
**Rating:** 4
**Confidence:** 4

**Summary:**

The paper proposes a classification of erasure methods into two categories:

•	Guidance-based erasure methods, which redirect the model’s generation process away from a target concept, while the to-be-erased concept’s knowledge remains intact within the model’s parameters.

•	Destruction-based erasure methods, which fundamentally suppress or eliminate the underlying knowledge of a concept from the model, making the concept theoretically irrecoverable.

Another contribution of the paper is an evaluation framework composed of four distinct evaluation methods:

•	Optimization-based Probing: Employs previously proposed techniques like Textual Inversion or adversarial attacks to search for adversarial prompts that recover the erased concept.

•	In-context Probing: A new evaluation method proposed by the authors. The idea is to provide visual cues—such as inpainting masked images or completing unfinished intermediate diffusion generations—as inputs (with visual hints) to test whether the erased concept can still be generated.

•	Training-Free Trajectory Probing: Another new evaluation method, which manipulates the diffusion process by adding noise to intermediate steps. As described in Appendix B, this is done by modifying the scaling factor $\eta$ in the random noise component in DDIM.

•	Dynamic Concept Tracing: Measures the distance between CLIP embeddings of images generated by the original model and the erased one (Section 3.4, Fig. 5). The authors claim that “Gradient Ascent and Task Vector—two destruction-based methods”—tend to generate more consistent replacements for the erased concepts, while guidance-based methods like ESD produce more varied outputs. However, the illustration in Fig. 6 and results in Fig. 5 seem to suggest the opposite. More specifically, in rows 2 and 3, the outputs from ESD converge stably to "a man," while rows 1 and 4 exhibit increasingly varied outputs. In Fig. 5, the centroid deviation for GA and TV increases over time, whereas that for ESD converges, which also aligns with the visualizations in Fig. 6.

**Questions:**

-	While I like the idea of in-context examples (Section 3.2), I couldn't understand how this module is implemented from the description in the paper. Specifically, in Appendix C.5.4 (Inpainting), it is mentioned that “the inpainting pipeline was based on SD v1.5 and implemented via SDInpaintPipeline.” Does this mean the authors applied unlearning to the inpainting model, rather than to the standard text-to-image SD model?

-	I don’t understand the discussion in Section 3.4 (Dynamic Concept Tracing). The authors state that “Gradient Ascent and Task Vector—two destruction-based methods—tend to generate more consistent replacements for the erased concepts, while guidance-based methods like ESD produce more varied outputs.” However, Fig. 6 and Fig. 5 seem to indicate the opposite. In rows 2 and 3, the outputs from ESD stably converge to “a man,” while rows 1 and 4 exhibit more variation. In Fig. 5, the centroid deviation of GA and TV increases over time, whereas that of ESD decreases. These results seem inconsistent with the authors' claims.

-	What is the target concept used in the experiments in Section 3.4 for the ESD-x and ESD-u methods? In my experience with ESD, the replacement output is strongly influenced by the choice of target concept, and typically will not default to “a man” unless that is explicitly defined as the target in the objective function.

-	In Training-Free Trajectory Probing, the authors propose adding noise to the intermediate latent $x$. What would happen if noise were added to the guidance signal instead (e.g., by perturbing the conditional signal $\tau(y) + \text{noise}$ or the classifier-free guidance signal $(\epsilon(\tau(y)) - \epsilon(\tau(\text{null})))$)?

**Ethical Concerns:**

["NO or VERY MINOR ethics concerns only"]

**Final Justification:**

in my opinion, this paper still offers valuable contributions that are relevant and would be of interest to the NeurIPS community:

While the current definitions of the two unlearning mechanisms are not yet rigorous, I believe that the conceptual proposal and awareness of these two paradigms are meaningful and necessary. Although I cannot pre-judge the final revision, I trust in the authors’ ability to refine and improve the presentation.

Unlike the Inpainting Probe, the proposed Diffusion Completion (Section 3.2) and Noise-Based Probing (Section 3.3) methods appear to be more general and scalable. I consider these to be interesting directions for evaluating unlearning in diffusion models.

In conclusion, I would like to keep my rating and lean towards accepting this paper.

**Limitations:**

Yes

**Paper Formatting Concerns:**

No concerns

**Quality:**

2

**Strengths And Weaknesses:**

**Strengths**

•  The paper is well written.

•  I particularly appreciate the idea of using in-context examples (Section 3.2) to provide visual cues for probing the generation capabilities or memory of the erased concepts. Overall, the Section 3 is well written and provide a comprehensive evaluation framework for evaluating concept unlearning.

**Weaknesses**

- Figure 1, while intuitive, may be misleading to me. Specifically, it is unclear or unconvincing why the redirection arrow (black arrow in the middle and right subfigures) should necessarily lead to a nearby concept. This redirection only makes sense if the erasure method—like ESD or UCE—explicitly forces the to-be-erased concept toward a targeted nearby concept, as done in [1], i.e., $\epsilon(c_e) \rightarrow \epsilon(c_t)$ where $c_t \in B(c_e)$. In standard erasure methods like ESD, UCE, RECE, or STEREO, where the target concept is not explicitly defined and where general or neutral concepts (e.g., “a photo,” an empty string “ ”, or something like “man” or “human”) are used instead, there is no guarantee that the redirected concept will be semantically nearby.

[1] Fantastic Targets for Concept Erasure in Diffusion Models and Where to Find Them

- The use of Textual Inversion (TI) can be highly sensitive to the amount and type of fine-tuning data. With a small number of reference images, TI may not perform well. However, with enough reference images, TI can learn personal or specific concepts—even if those have been erased.

- Another concern is the definition of “concept”, which can be ambiguous. For example, a concept like “an astronaut monkey” may not appear in the training data but can still be generated by the model based on the core concepts of “astronaut” and “monkey.” Similarly, abstract or nuanced concepts such as “nudity” or artistic styles, though complex, may still be reconstructed from more basic elements (e.g., “human body,” “human skin”). Thus, I believe that achieving complete destruction (as claimed in destruction-based removal) may be unrealistic. This is supported by the results in Table 1: GA, labeled as a destruction-based method, can still regenerate the erased concept using Textual Inversion. I suspect that with a large enough reference dataset, TI would recover the concept even more effectively. Therefore, while the modeling distinction between guidance-based and destruction-based removal is conceptually useful, I believe most current methods—including those labeled as destruction-based—are closer to guidance-based in practice.

---

> ### Author Rebuttal · Authors · 2025-07-29
>
> We thank Reviewer r56k for their positive assessment of our work and for highlighting the clarity of our writing and the novelty of the in-context probing evaluation. We are especially glad that you found Section 3's evaluation framework comprehensive and well motivated.
>
>  **“it is unclear or unconvincing why the redirection arrow (black arrow in the middle and right subfigures) should necessarily lead to a nearby concept.”**
>
> Thank you for your question. The arrows on Fig.1 come to illustrate the idea of an alternative generation, “chosen” by a model that underwent a destruction based erasure. As shown in Fig.5 and Fig.6, the alternative generation leads to a nearby concept to the original one when the erasure strength is small (erasure strength being the method “strength” parameter, e.g., fine-tuning epochs). Even and especially in models that do not explicitly define “neutral concepts”. Yet, as the strength parameter increases, the alternative generation is indeed driven to be further away from the original concept.
>
> In any case, we would like to clarify Fig.1 comes to illustrate the idea of the alternative concept distance from the original one, and not necessarily to claim it is nearby (or faraway) for any specific method. We have now further clarified this in the caption.
>
>
>  **“The use of Textual Inversion (TI) can be highly sensitive to the amount and type of fine-tuning data”**
>
> We thank the reviewer for their comment. We report below further experiments to explore the sensitivity of the Textual inversion attacks on our chosen erased models, by training TI embeddings on 1, 10, 100, and 300 images for five representative erasure methods.
>
> With limited reference images, TI may indeed struggle to recover erased concepts in some cases And as shown in the table below, increasing the number of TI training images significantly improves both CLIP scores and classification accuracy for certain models, indicating that erased concepts can often be relearned through TI if sufficient data is provided.
>
> Notably, models like UCE and ESD-X showed substantial increases in classification accuracy as the number of used images increased, whereas methods such as GA and TV remained more resistant to TI attacks.
>
> | | GA | UCE | ESD-X | ESD-U | TV |
> |------------------------|-------|-------|-------|-------|-------|
> | **1 Image** | | | | | |
> | CLIP | **20.5** | 26.3 | 25.9 | 24.7 | 22.0 |
> | Class Acc. (%) | **0.0** | 35.0 | 30.0 | 10.0 | 0.0 |
> | **10 Images** | | | | | |
> | CLIP | **21.8** | 28.6 | 28.9 | 26.5 | 23.5 |
> | Class Acc. (%) | **0.2** | 58.0 | 51.0 | 20.0 | 1.0 |
> | **100 Images** | | | | | |
> | CLIP | **22.7** | 30.7 | 30.6 | 28.0 | 25.1 |
> | Class Acc. (%) | **0.6** | 71.2 | 65.9 | 31.8 | 6.2 |
> | **300 Images** | | | | | |
> | CLIP | **23.0** | 30.9 | 30.3 | 30.2 | 26.4 |
> | Class Acc. (%) | **1.0** | 72.3 | 67.3 | 33.9 | 8.5 |
>
>   **"I believe that achieving complete destruction (as claimed in destruction-based removal) may be unrealistic."**
>
> We agree that achieving complete destruction of a concept may be difficult in practice, or even impossible particularly when powerful recovery strategies like Textual Inversion (TI).
>
> Our intention is not to claim that current methods fully realize the destruction based ideal case, but rather to organize the space of existing approaches and motivate future work that pushes closer toward true destruction.
>
> That said, we appreciate the reviewer’s broader point: while stronger methods such as GA are better at thoroughly removing concepts, we acknowledge they may still not achieve a complete destruction of the concept. While developing fully robust methods may remain an open challenge, our paper aims to examine existing approaches and motivate future work that pushes closer toward true destruction.
>
>   **“Does this mean the authors applied unlearning to the inpainting model, rather than to the standard text-to-image SD model?”**
>
> Thank you for the question. The unlearning was applied only to SD v1.4 models. We found that using a base SD v1.5 inpainting pipeline with an SD v1.4 erased UNet produced more stable and visually coherent outputs than using the SD v1.4 inpainting pipeline.
>
>
> **Regarding Figure.6: Consistent generations**
>
> In line 196, with the claim: “Gradient Ascent and Task Vector often converge on a consistent alternative concept”, our intention was to divert the reader’s attention to the evolution of the concept through the unlearning process. This can be seen in row 2, GA’s outputs throughout the unlearning training smoothly converges to the same ‘mushy’ concept whereas ESD has a more ‘erratic’ convergence to its final concept. We have clarified this better in our text.
>
> **Regarding Figure.5: Centroid plots**
>
> In Fig. 5, we do not plot standard deviation across generations. Rather, we show the average distance between the original and alternative generation embeddings. GA and TV show steady increases, indicating a smooth drift away from the original concept. ESD-U and ESD-X, however, exhibit sharp spikes in this metric: corresponding to the abrupt visual transitions seen in Fig. 6.
> We will clarify it in the final version of the manuscript and add the distribution images to show the variations of each model.
>
>
>
>
> **What is the target concept used in the experiments in Section 3.4 for the ESD-x and ESD-u methods?**
>
> We thank the reviewer for their observation and would like to clarify. The target concept used for these methods was "vincent van gogh self portrait." Accordingly, the trajectories converged to "a man", diverging away from Van Gogh as the erasure progressed. In accordance with the reviewer's comment about the target concept having a strong influence on the replacement output, we have run ESD-u and ESD-x with the target concept "a painting in the style of van gogh," and the resultant trajectories converged to images of a painting.
>
>
> **“What would happen if noise were added to the guidance signal instead”**
>
> We thank the reviewer for their insightful suggestion. To investigate this, we conducted a detailed analysis of perturbing different components of the classifier-free guidance expression:
>
> $$
> \epsilon_{\text{guided}} = \epsilon_u + s \cdot (\epsilon_c - \epsilon_u)
> $$
>
> We find that perturbing the conditional prediction ($\epsilon_c \rightarrow \epsilon_c + \sigma z$) or the guidance vector ($( \epsilon_c - \epsilon_u ) \rightarrow ( \epsilon_c - \epsilon_u + \sigma z )$) results in the same final expression:
>
> $$
> \epsilon_{\text{guided}} = \epsilon_u + s \cdot (\epsilon_c - \epsilon_u) + s \sigma z
> $$
>
> However, perturbing the unconditional prediction ($\epsilon_u \rightarrow \epsilon_u + \sigma z$) yields a subtly different form:
>
> $$
> \epsilon_{\text{guided}} = (\epsilon_u + \sigma z) + s \cdot (\epsilon_c - (\epsilon_u + \sigma z)) \\
> = \epsilon_u + s \cdot (\epsilon_c - \epsilon_u) + (1 - s) \sigma z
> $$
>
> Notably, the noise term here is scaled by $(1 - s)$, which becomes negative when $s > 1$. This introduces an anti-aligned perturbation relative to the guidance direction, effectively suppressing or counteracting the text guidance. To explore this effect, we now include both conditional and unconditional perturbation variants in the table below:
>
> |                        | **ESD-u** | **ESD-x** | **UCE** |
> |------------------------|----------|----------|---------|
> | **CLIP Score (↑)**     |          |          |         |
> | Unconditional-Signal   | 26.90    | 27.35    | 27.10   |
> | Conditional-Signal     | 25.80    | 29.10    | 29.05   |
> | Noising Probe          | **27.99**| **30.56**| **30.65** |
> | **Top-1 Accuracy (%) (↑)** |      |          |         |
> | Unconditional-Signal   | 25.90    | 28.50    | 19.35   |
> | Conditional-Signal     | 23.62    | 28.37    | 16.61   |
> | Noising Probe          | **27.70**| **30.70**| **21.90** |
>
>
> Interestingly, we observed that perturbing the unconditional signal occasionally led to out-of-distribution generations, often producing semantically ambiguous or hallucinatory outputs. Despite this, concepts that were difficult to recover using the standard noising probe (like English Springer Spaniel), were better recovered using the unconditional perturbation. This suggests that such perturbations may expose residual concept traces in the models learned prior that are not typically activated during normal guidance.
>
> We sincerely thank the reviewer for their thoughtful suggestions and questions. We will revise the final version of the manuscript to improve clarity, and we welcome the opportunity to continue this discussion.

---

> > ### Comment · Reviewer_r56k · 2025-08-05
> > **Response to the rebuttal**
> >
> > I thank the authors for their efforts in addressing my concerns—some of which have been clarified. However, I would appreciate further elaboration on the following points:
> >
> > **Definition of Mechanisms (Guidance-based vs. Destruction-based):**
> >
> > While the distinction between guidance-based and destruction-based removal is intuitive, I believe these concepts require more rigorous and precise definitions to avoid potential misinterpretation—particularly for the destruction-based category. For instance, in the illustration in Figure 1 (Z-axis), what exactly is being visualized? Is it the likelihood of the target concept in terms of P(x|c), P(c|x), P(c|G(c)), or simply P(c)? A clearer formalization would enhance the validity and reproducibility of the framework.
> >
> > **Potentially Misleading Conclusion:**
> >
> > In Figure 5 and elsewhere, the paper categorizes GA and TV as destruction-based methods. However, this conclusion seems to rely heavily on their robustness to specific recovery attacks. This may not necessarily imply that the methods fundamentally remove concepts in a destruction-based manner. The situation is reminiscent of the adversarial robustness literature—particularly the findings in “Obfuscated Gradients Give a False Sense of Security: Circumventing Defenses to Adversarial Examples.” As noted in that work (and raised by Reviewer H6xi), some defenses appear robust only because they are evaluated under weak attacks. Similarly, your own experiments (e.g., Figure 7) demonstrate that GA can still regenerate the erased concept under more effective recovery probes. This suggests that GA may not satisfy the criteria for true destruction-based removal.
> >
> > **Figure 5 – Long-Term Behavior:**
> >
> > Thank you for the clarification regarding Figure 5. I am curious whether GA and TV eventually converge over a longer observation window. In Figure 6, convergence is shown clearly, and it would be interesting to know whether a similar pattern is observed for GA and TV under extended probing.
> >
> > **Inpainting-based Probe with an Erased UNet:**
> >
> > Regarding the question, “Does this mean the authors applied unlearning to the inpainting model, rather than to the standard text-to-image SD model?”—the rebuttal states that a base SD v1.5 inpainting pipeline was used in combination with an SD v1.4 erased UNet. My understanding (based on Hugging Face documentation) is that inpainting models are typically obtained via a fine-tuning phase from a pretrained SD model. If so, how exactly does the inpainting-based probe function when combined with a UNet from a text-to-image model that has undergone unlearning? Some clarification about the compatibility and mechanics of this integration would be very helpful.
> >
> > **Perturbing the Conditional Signal τ(y):**
> >
> > Thank you for including the additional experiments on perturbing the guidance signal. In my original question, I was also interested in what would happen if noise were added directly to the conditional signal—i.e., perturbing the textual encoding τ(y) → τ(y) + noise, where τ(·) is the text encoder and y is the input prompt. Could the authors provide any empirical results or theoretical discussion on how such perturbation might affect unlearning behavior or recovery performance?
> >
> > Overall while I like the set of evaluation in Section 2, I am still concerned about the rigorous of the two unlearning mechanisms and the misleading that it might bring.

---

> ### Author Response · Authors · 2025-08-06
>
> Thank you for engaging in the discussion! Please find our further clarifications below:
>
>
> **Definition of Mechanisms**
>
> Figure 1 (Z-axis) indeed visualizes P(x). While serving illustrative purposes, we fully agree that this should be made clearer and appreciate the reviewer’s attention to this point.
>
>
> **Potentially Misleading Conclusion**
>
>
> Thank you for this opportunity to clarify. We refer to methods such as GA as “destruction-based” as we claim the tend to edit the model unconditional likelihood ($P(x)$) rather than mostly preserving $P(x)$ and editing $P(x|c)$. Yet, in practice, $P(x)$ for the target concept is reduced albeit not necessarily to zero.
>
> We will make it clearer in the final version of the manuscript.
>
>
> **Figure 5 – Long-Term Behavior**
>
>
> We would like to clarify that for ever-increasing values of edit strengths (e.g., number of epochs for GA or vector magnitude for TV), as the target concept becomes more thoroughly erased, the ability of the model to generate unrelated concepts is destroyed as well.
> Yet, per the reviewer request, we are happy to report the consistency of alternative-generation for larger edit strength:
> | **Method** | **Edit Strength**    | **CLIP Centroid Distance** |
> |------------|----------------|-------------------------------|
> | GA         | 50 steps       | 4.42                          |
> | GA         | 100 steps      | 4.47                          |
> | GA         | 200 steps      | 4.47                          |
> | TaskVec    | α = 5          | 4.91                          |
> | TaskVec    | α = 10         | 5.16                          |
> | TaskVec    | α = 25         | 5.04                          |
>
> We observed that as edit strengths increased beyond typical thresholds, the CLIP centroid deviations rose gradually before plateauing, which aligns with the expected behavior of GA and TV. However, the generated images at these extreme edit levels often appeared broken or degraded, reflecting behavior well beyond that of standard erasure methods.
>
>
> **Inpainting-based Probe with an Erased UNet:**
>
>
> Thank you for this question. To clarify the mechanics: we used the Hugging Face StableDiffusionInpaintPipeline as a convenient wrapper to handle the masked latent processing, but we did not use a specifically fine-tuned inpainting model. The inpainting pipeline essentially handles the mechanics of: (1) encoding the masked image into latent space, (2) adding noise to the masked regions, and (3) running the diffusion process where our erased UNet must complete the masked areas. While Hugging Face recommends using specifically fine-tuned models for inpainting, their documentation also states that standard models like SD 1.5 can be used effectively. Since SD 1.4 and SD 1.5 share identical UNet architectures, this replacement ensures architectural compatibility and consistency with the rest of our experiments.
>
>
> Our goal was not to unlearn an inpainting model, but to use the inpainting setup as a unique probe for evaluating concept erasure in a “base” diffusion model.
>
>
> **Perturbing the Conditional Signal τ(y):**
>
>
> Thank you for the clarification! We conducted a follow-up experiment where we added a Gaussian noise vector to the text embedding (conditional) signal across a range of 25 noise strengths (0.0 to 5.0) and across four concepts (“church”, “airliner”, “golf ball”, “Van Gogh”). The results are shown below:
>
>
> Conditional Signal Noising – Evaluation Table
>
> | **Metric**              | **ESD-x** | **ESD-u** | **UCE** | **GA**  | **Stereo** |
> |-------------------------|:--------:|:--------:|:------:|:------:|:--------:|
> | **CLIP Score**          | 26.12    | 24.80    | 27.01  | 25.74  | 22.99    |
> | **Classification Acc.** | 17.00%   | 1.67%    | 14.00% | 0.00%  | 0.00%    |
>
>
> We found that methods focusing on editing the cross-attention layers (such as UCE and ESD-X, which directly interact with the prompt encoding) were particularly vulnerable to this type of perturbation. We also found that approximately 80% of successful generations occurred at higher noise levels (ε > 3.0), but beyond ε = 5.0, the images became less recognizable.
>
>
> Thank you again for your insightful suggestions and follow-up comments. We would be glad to continue the discussion further during the remainder of the review period.

---

> > ### Comment · Reviewer_r56k · 2025-08-07
> > **Further comments**
> >
> > I thank the authors for the additional clarifications and experiments. I truly appreciate and acknowledge the effort you have put into addressing the concerns. However, I still have some remaining points of concern, detailed below:
> >
> > **Definition of the Two Unlearning Mechanisms: Guidance-Based Avoidance and Destruction-Based Removal**
> >
> > The paper lacks a rigorous and formal definition of these two unlearning mechanisms, which may lead to misleading conclusions. For instance, the claim that GA or TV are destruction-based methods appears insufficiently justified. Similarly, the caption in Figure 1—“forcing the model to a nearby concept when prompted with the target concept”—raises questions. In Figure 6, the concept trajectory for GA leads into severely distorted images. In this case, what exactly qualifies as a “nearby concept”?
> >
> > **Caution with Conclusions on Unlearning Robustness (i.e., Whether Target Knowledge Persists)**
> >
> > In Section 3.1, the authors evaluate several unlearning methods using Optimization-based Probing (Textual Inversion and UnlearnDiffAttack), and conclude that “GA, TV, and STEREO exhibit thorough removal of the erased concept,” as indicated by low CLIP similarity and classification accuracies across both probes. I would caution against such a strong conclusion. As highlighted in the paper “Obfuscated Gradients Give a False Sense of Security: Circumventing Defenses to Adversarial Examples”, robustness observed under weak or poorly designed attacks may give a false sense of security. In this context, good performance against TI and UnlearnDiffAttack does not necessarily indicate complete concept erasure—it may simply reflect the weakness of these probing methods. I suggest rephrasing the conclusion to something like:
> > “GA, TV, and STEREO exhibit robustness against the two optimization-based probes, suggesting a certain level of concept removal under these specific attack settings.”
> >
> > **Limitation of the Inpainting-Based Probe**
> >
> > While I appreciate the novelty of the proposed “In-context Probing,” I still have concerns about the scalability and generalizability of this method. As clarified by the authors, the approach involves unlearning on the standard SD v1.4 T2I model and plugging the resulting UNet into the SD v1.5 Inpainting Pipeline. Thanks to the compatibility between the two versions, no further fine-tuning is required. However, this compatibility may not hold in general—what happens if the unlearned model is based on a different version that is incompatible with existing inpainting pipelines? The method’s general applicability in such cases remains unclear.
> >
> > **Overall Assessment**
> >
> > However, in my opinion, this paper still offers valuable contributions that are relevant and would be of interest to the NeurIPS community:
> >
> > - While the current definitions of the two unlearning mechanisms are not yet rigorous, I believe that the conceptual proposal and awareness of these two paradigms are meaningful and necessary. Although I cannot pre-judge the final revision, I trust in the authors’ ability to refine and improve the presentation.
> >
> > - Unlike the Inpainting Probe, the proposed Diffusion Completion (Section 3.2) and Noise-Based Probing (Section 3.3) methods appear to be more general and scalable. I consider these to be interesting directions for evaluating unlearning in diffusion models.
> >
> > In conclusion, I would like to keep my rating and lean towards accepting this paper.

---

> ### Author Response · Authors · 2025-08-08
>
> We would like to thank the reviewer for the thoughtful discussion and for recognizing the contributions of our work.
>
> We agree it is important to be more cautious and precise in our claims about robustness, and we will revise the manuscript accordingly. We will ensure that our language clearly distinguishes between our conceptual categorization and empirical claims about specific methods, which may not align perfectly with the categorization. Additionally, we will use more precise language to describe alternative generations and the inpainting process.
>
> Thank you again for your constructive feedback!

---

### Official Review · Reviewer_jc67 · 2025-07-03

**Clarity:** 3
**Significance:** 3
**Originality:** 3
**Rating:** 5
**Confidence:** 4

**Summary:**

This paper investigates whether concept erasure *actually* removes the latent concepts or merely suppress their likelihood of generation through gradient redirection during diffusion. This is formalized as (1) guidance-based avoidance and  (2) destruction-based removal. To answer the four RQs, the authors compile an evaluation suite comprising optimization-based probing (textual inversion, adversarial attacks), in-context probing (inpainting, diffusion completion), training-free trajectory probing (noise injection), and dynamic concept tracing during erasure. This is applied to seven seven methods and 13 concepts — 10 objects and 3 artistic  styles — highlighting that these methods are usually robust on one of the defined types of erasure but struggle on the other.

**Questions:**

1. Method Categorization. How do you determine ig a method is guidance- or destruction-based a priori, rather than post-hoc based on empirical performance? Could you provide more principled criteria or theoretical analysis to validate this categorization?

2. Scalability to Harmful Content: How would your framework perform on concepts that are often used as motivation for the existance of concept erasure (e.g., explicit content or copyrighted material)?

**Ethical Concerns:**

["NO or VERY MINOR ethics concerns only"]

**Final Justification:**

As noted below, the vast majority of my concerns were fully resolved. One concern partially remained, but I felt like given the effort exhibited by the authors in the response, and technical rigor of the existing experiments, I am confident moving my rating to an Accept with the string recommendation for reframing the significance of some of the findings.

**Limitations:**

Yes, the authors adequately address limitations in Section 5, acknowledging the restricted concept scope, causality issues in concept erasure, and the potential need for additional conceptual models. The broader impact section (app A) appropriately discusses both positive applications and potential misuse concerns. However, the authors could strengthen this by discussing the computational overhead of their evaluation framework and potential limitations in scalability to production systems.

**Paper Formatting Concerns:**

Capitalization in the references (e.g., “Lora”, “Unpacking sdxl turbo: Inte…”)

**Quality:**

3

**Strengths And Weaknesses:**

Strengths:
1. New Conceptual Framework. The distinction between guidance-based avoidance and destruction-based remova is a really valuable theoretical grounding for the community and future work in this direction.
2. Comprehensive Evaluation Suite.
3. Practical Insights. The finding that methods, which may appear quite robust under adversarial attacks can still be vulnerable to context-based probes (tab 2, fig 3), has important implications for deployment of erasing methods.
Weaknesses:
1. Limited Scope of Concepts. The evals cover 13 concepts (10 objects and 3 art styles), which may not generalize to some more complex concepts or larger domains of objects. The authors acknowledge this limitation (Section 5, lines 267-269). Though, it’s not clear to me why, instead of so many single objects, the authors did not choose to explore this.
2. Validation of the Theoretical Framework: While the two modes of erasure proposed here seem intuitive, the paper would benefit from more support of this categorization. The assignment of methods to categories appears somewhat post-hoc based on empirical performance rather than principled analysis of the underlying mechanisms.
3. Trade-off Analysis. While the paper mentions the trade-off between erasure thoroughness and preservation of unrelated concepts (lines 45-46, 123-124), this aspect could, perhaps, benefit from a more quantitative analysis.
4. Noise-Based Probing Limitations: The theoretical justification for the noise-based probe (app B) relies on some strong assumptions (partial latent retention and specific SDE formulations). The proof sketch appears limited to me compared to other probing methods (tab 3).
5. Prior Work. The issues with existing concept erasure methods have been identified before, and I would hence suggest the following citations: https://arxiv.org/pdf/2502.13989v1 https://dl.acm.org/doi/10.1145/3635636.3664268 https://arxiv.org/abs/2502.14896

---

> ### Author Rebuttal · Authors · 2025-07-29
>
> We thank the reviewer for acknowledging the value of the guidance-based avoidance and destruction-based removal concept erasing framework. As you point out, results emphasize the importance of testing erasure methods across multiple probing types.
>
>   **“Limited Scope of Concepts”, “Scalability to Harmful Content”**
>
> We thank the reviewer for raising this point and agree that expanding to other concepts would strengthen our results. In the initial study, we deliberately chose a focused set to control for confounding factors such as prompt ambiguity, visual diversity, and domain shifts.
>
> To assess the scalability of our framework to safety-relevant domains, we evaluated it on NSFW content, specifically nudity. We use 100 prompts from the I2P dataset and evaluate nudity scores using the NudeNet classifier.
> As shown in the table below, our framework successfully captures differences in guidance vs destruction-based erasing behavior across the methods. Interestingly, the Noise-Based probe (Sec.3.3) was surprisingly effective at revealing residual NSFW features in erased models. For example, using this probe, ESD-u's classification rate increased from 23% to 79%. This reinforces the utility of our multi-probe evaluation in these settings as well.
>
> | | SD | UCE | ESD-U | RECE | STEREO | GA | TV |
> |---------------------------------|-------|-------|-------|-------|--------|-------|-------|
> | **Ground Truth** | | | | | | | |
> | Avg. Nudenet Score | 0.57 | 0.22 | 0.11 | **0.05** | 0.06 | 0.10 | 0.07 |
> | Classification (%) | 83.0 | 39.0 | 23.0 | **12.0** | 18.0 | 28.0 | 21.0 |
> | **Interference Probe** | | | | | | | |
> | Avg. CLIP Score | 31.20 | 29.66 | **30.50** | 29.97 | 28.61 | 27.23 | 29.92 |
> | Classification (%) | 95.2 | 72.0 | **76.0** | 75.2 | 65.0 | 45.0 | 70.0 |
> | **Textual Inversion** | | | | | | | |
> | Avg. Score | 0.81 | 0.73 | 0.75 | 0.69 | **0.61** | 0.72 | 0.65 |
> | Classification (%) | 99.8 | 87.4 | 88.1 | 75.2 | **28.5** | 29.0 | 54.0 |
> | **UDA** | | | | | | | |
> | Avg. Score | 0.70 | 0.66 | 0.71 | 0.60 | **0.55** | 0.69 | 0.61 |
> | Classification (%) | 95.5 | 74.3 | 62.0 | 69.1 | **41.0** | 44.3 | 57.5 |
> | **Inpainting** | | | | | | | |
> | Avg. Nudenet Score | 0.45 | 0.30 | 0.31 | 0.29 | **0.28** | 0.29 | 0.33 |
> | Classification (%) | 72.0 | 54.0 | 53.0 | 57.0 | **51.0** | 52.0 | 60.0 |
> | **Diffusion Completion (t=5)** | | | | | | | |
> | Avg. Score | 0.57 | 0.68 | 0.68 | 0.71 | 0.66 | **0.59** | 0.69 |
> | Classification (%) | 83.0 | 75.6 | 74.9 | 77.2 | 65.8 | **56.4** | 75.1 |
> | **Noise-Based Probe** | | | | | | | |
> | Avg. Score | 0.72 | 0.53 | **0.70** | 0.52 | **0.48** | 0.62 | 0.51 |
> | Classification (%) | 99.7 | 73.0 | **79.0** | 63.0 | **37.0** | 42.0 | 72.0 |
>
>
>
> **“more support of this categorization”**
>
> Our motivation behind guidance vs destruction categorization is based on the resultant distribution of the erased concept post erasure. With guidance based methods, the unlearning algorithm pushes the conditional probability $P(X | c)$, where $c$ is the erased concept, to another known distribution. For example, with UCE we can write the erased distribution $P(X | c*)$ where $c*$ is the anchor concept, which comes from the following loss function:
>
> $$
> \mathcal{L}(W) = \sum_{c_i \in E} \left\| W c_i - v_i^* \right\|^2 + \sum_{c_j \in P} \left\| W c_j - W_{\text{old}} c_j \right\|^2
> $$
>
> Specifically, UCE learns a new attention projection matrix \\( W \\) that maps erased concept prompts \\( c_i \\) to substitute vectors \\( v^*_i \\), while preserving the responses to protected concepts \\( c_j \\).
>
> On the other hand, other methods (e.g., Gradient Ascent, TV) use alternative objectives such as increasing the diffusion loss. Even if we suspect that the erasing mechanism removes explicit knowledge of the target concept from the model, the resultant concept distribution is difficult to explicitly compute and categorize. To tackle this challenge, we have thus proposed our suite of evaluations described in the paper as an empirical tool for analyzing erased models.
>
> **“Trade-off … more quantitative analysis”**
>
> We thank the reviewer for their suggestion. To address it, we define a Combined Method Score, which is an average of the method’s normalized erasing performance (across the different probes), unrelated concept preservation, and FID score. We calculated the Fréchet Inception Distance (FID) on the MS-COCO30k dataset for each model to assess the "quality" of images generated by these erased models compared to the original model as yet another axis of evaluation.
>
> This Combined Method Score reveals several interesting trends: STEREO outperforms ESD-U, the method it builds upon; and RECE similarly improves on its base method, UCE. Although both GA and STEREO achieve near-perfect erasure scores, their overall performance diverges: STEREO ranks highest overall, while GA ranks lowest. This disparity is driven by GA's poor FID and preservation scores, which offset its strong erasure.
>
> |**Rank**|**Method**|**Erasure**|**FID (↓)**|**Preservation**|**Model Score**|
> |---|---|---|---|---|---|
> |1|Stereo|**0.99**|19.5|0.74|**0.81**|
> |2|ESD-U|0.76|17.4|0.86|0.80|
> |3|TaskVec|0.90|18.8|0.79|0.80|
> |4|RECE|0.69|17.5|0.87|0.78|
> |5|UCE|0.60|**16.5**|**0.90**|0.76|
> |6|ESD-X|0.63|17.1|0.87|0.75|
> |7|GA|0.97|21.2|0.74|0.73|
> |8|Base (SD 1.4)|0.00|16.0|1.00|0.50|
>
>
>
> **“theoretical justification for the noise-based probe (app B) relies on some strong assumptions”**
>
> We thank the reviewer for raising this concern. While our analysis does rely on assumptions such as partial latent retention and the SDE-based formulation of diffusion, we emphasize that this is not intended as a formal proof characterizing the Noise-Based probe. In particular, we used it as a motivation for why introducing noise into the diffusion process would expand the space of “useful” trajectories for recovering images of erased concepts. We have now made this clearer in the revised manuscript.
>
> **“Prior Work”**
>
> Thank you for these helpful suggestions. We agree these works provide meaning context to our work and we incorporated these citations into the manuscript.
>
> **“discussing the computational overhead … potential limitations in scalability”**
>
> Thank you for this thoughtful suggestion. In Appendix C.2, we report that running the full evaluation suite (spanning 7 erasure methods and 13 concepts across all 6 probes) took approximately two weeks on two NVIDIA A6000 GPUs, or roughly one day per method per concept. We agree that as production systems begin to scale verification across hundreds of concepts and models, this level of computational cost could pose practical limitations.
>
> While our framework is designed for comprehensive academic benchmarking, we recognize that production settings may benefit from more lightweight diagnostics. However, several of our introduced probes, such as the noise-based probe and inpainting-based context probe, are training-free and are in fact faster to run than other approaches.
>
> To illustrate, we provide a rough breakdown of compute cost by probe (as a percentage of total runtime):
> 1.  Textual Inversion: 39%
> 2.  UnlearnDiffAtk (UDA): 37%
> 3.  Inpainting: 3%
> 4.  Diffusion Completion: 4%
> 5.  Noise-Based Probe: 7%
> 6.  Dynamic Concept Tracing: 10%
>
> Taken together, the training-free probes (inpainting, noise, diffusion completion, and concept tracing) require less than 15% of the total runtime.
>
> Thank you again for your excellent comments. We will revise the final version of the manuscript to include your kind suggestions. We are happy to continue the discussion at any time until the end of the discussion period. Thank you!

---

> > ### Comment · Reviewer_jc67 · 2025-08-06
> >
> > Thanks for carefully responding to all of my points. Except for the first point on limited scope of concepts, I see all of the responses as satisfactory -- either providing new analysis that fixes/clarifies previous shortcomings, or explaining original experiments.
> >
> > On the first point: I agree that the existing experiments are very extensive and informative in terms of NSFW content. I do think one can make some predictions or connections to performance beyond NSFW, but without additional experiments, I would be careful with the extent and strength of these experiments.
> >
> > That being said, since all of my other concerns have been resolved, and the authors have demonstrated a level of technical rigor in their experiments, I will move my assessment from 'Borderline Accept' to 'Accept'. I will, nonetheless, suggest that the framing and scope of statements is an adjusted to be more bound to the domain of NSFW content. You should make clear distinction between presenting results and findings that are directly demonstrated in the data you have, and discussion weighing broader impact and applicability.

---

> > > ### Author Response · Authors · 2025-08-06
> > >
> > > We would like to thank the reviewer for their thoughtful discussion and for expressing an even more positive view of our work.
> > >
> > > We will make sure to adjust the scope of our statements to better align with the experimental evaluation that includes NSFW content, daily objects, and art styles, but no necessarily all concepts of interest.
> > >
> > > Thank you once again for your excellent comments!

---

### Author Response · Authors · 2025-08-08

We thank all the reviewers for their valuable feedback!

Our work aims to evaluate whether unlearning methods for diffusion models maintain knowledge about supposedly erased concepts. Breaking away from previous research, we explore four different types of probes to provide distinct forms of conditioning for concept recovery. We appreciate that the reviewers found our suggested work valuable for the community and providing practical insights (*jc67*), fundamental and well motivated (*Bk4k*), and intuitive (*H6xi*). The reviewers also acknowledge our comprehensive evaluation (*jc67, r56k, Bk4k*) and recognize our paper as well written (*r56k, Bk4k*).

Following the discussion with the reviewers, we highlight the following improvements:

- We better explain our conceptual categorization and how it relates to our empirical evaluation.
- We extend our evaluation to include additional types of concepts.
- We provide additional results on the trade-off between erasure thoroughness and model preservation.

Once again, we sincerely thank the area chair and all reviewers!

---

### Decision · Program_Chairs · 2025-09-17

**Decision:**

Accept (poster)

**Comment:**

This paper attempts to address an important question that the concept unlearning community is grappling with across the board: are current algorithms actually removing information or simply suppressing them. This is a very complicated question and this paper makes some good first steps towards understanding this phenomenon. The paper starts by categorizing concept unlearning methods as suppression based or destruction based. It then provides four different probes to measure this distinction.

Reviewers appreciated the novelty and importance of the insights provided in this paper. Especially regarding how to better measure whether true unlearning has happened. The main weaknesses of this paper at first were the lack of rigour in the categorization of guidance based vs. destruction based erasure, the conclusions that could be drawn from the robustness experiments, and the computational costs from running these probes. The majority of reviewers were satisfied with the responses to these weaknesses with some remaining minor concerns. I strongly encourage the authors to focus on improving the clarity of the distinction between guidance and destruction based erasure and grounding it in more rigorous interpretations as suggested by 56k.

While one reviewer felt strongly about rejection, having gone through more of the paper myself and all of the reviews + rebuttals I am recommending acceptance. I believe that the insights in this paper and the proposed probes will be a good starting point for the community to better understand just how poorly our current unlearning algorithms are performing.